# α-Catenin levels determine direction of YAP/TAZ response to autophagy perturbation

Mariana Pavel [1,2,5], So Jung Park[1,3,5], Rebecca A. Frake[1], Sung Min Son[1,3], Marco M. Manni[1,3], Carla F. Bento[1], Maurizio Renna[1], Thomas Ricketts[1], Fiona M. Menzies[1], Radu Tanasa[4] & David C. Rubinsztein [1,3✉]

The factors regulating cellular identity are critical for understanding the transition from health to disease and responses to therapies. Recent literature suggests that autophagy compromise may cause opposite effects in different contexts by either activating or inhibiting YAP/TAZ co-transcriptional regulators of the Hippo pathway via unrelated mechanisms. Here, we confirm that autophagy perturbation in different cell types can cause opposite responses in growth-promoting oncogenic YAP/TAZ transcriptional signalling. These apparently contradictory responses can be resolved by a feedback loop where autophagy negatively regulates the levels of α-catenins, LC3-interacting proteins that inhibit YAP/TAZ, which, in turn, positively regulate autophagy. High basal levels of α-catenins enable autophagy induction to positively regulate YAP/TAZ, while low α-catenins cause YAP/TAZ activation upon autophagy inhibition. These data reveal how feedback loops enable post-transcriptional determination of cell identity and how levels of a single intermediary protein can dictate the direction of response to external or internal perturbations.

[1] Department of Medical Genetics, Cambridge Institute for Medical Research, Wellcome Trust/MRC Building, Cambridge, UK. [2] Department of Immunology, Grigore T. Popa University of Medicine and Pharmacy of Iasi, Iasi, Romania. [3] UK Dementia Research Institute, Cambridge Biomedical Campus, Cambridge, UK. [4] Department of Physics, Alexandru Ioan Cuza University of Iasi, Iasi, Romania. [5] These authors contributed equally: Mariana Pavel, So Jung Park. ✉email: dcr1000@cam.ac.uk

The identity and status of cells are commonly depicted using static "omics" profiles: epigenomic, transcriptomic, proteomic or metabolomic[1–4]. Changes in cellular identity play an important role in promoting diseases, including cancers, inflammatory conditions and neurodegeneration, and identifying the factors regulating these cell identity switches might provide novel therapeutic opportunities[5,6]. However, it is not well understood how such static hallmarks change over time and convert into dynamic responses to external or internal perturbations that ultimately determine cellular functions.

Macroautophagy (henceforth autophagy) is a conserved pathway where cytoplasmic contents are engulfed in double-membrane vesicles called autophagosomes, which ultimately fuse with lysosomes enabling degradation of their contents[7–10]. Autophagy plays important roles in physiology, including buffering against starvation, and regulates susceptibility to numerous diseases, like cancer, infections and neurodegeneration[11–16]. As autophagy modulators may prove beneficial for some diseases, it is critical to decipher the cellular responses they induce. For example, it is unclear whether autophagy inhibition is beneficial or deleterious in cancers[17], and some of the controversy in this area has resulted from the expectation that all cancers may respond similarly to autophagy modulation.

Since certain effectors of critical signalling pathways are autophagy substrates, we expected that autophagy inhibition in different cell types would cause signalling responses in the same directions but possibly with differing amplitudes according to cell type. Those various reaction amplitudes would most likely result either from different starting levels of the effector proteins or varying baseline autophagic flux in distinct cell types. However, recent literature suggests that autophagy compromise may cause opposite effects in different cell lines (e.g., either activating[18,19] or inhibiting[20] YAP/TAZ co-transcriptional regulators of the Hippo pathway) via unrelated mechanisms.

Here, we describe the surprising finding that autophagy inhibition causes divergent positive and negative responses in YAP/TAZ activity in different cell types. This results from different starting levels of a common key signalling effector, which acts as an autophagy substrate: α-catenin. This mechanism is further completed by the dynamic mutual regulation of YAP/TAZ by autophagy and of autophagy by YAP/TAZ. Thus, we are showing that YAP/TAZ activity is a cellular readout that is determined by two competing processes: one being represented by the ability of autophagy to directly degrade YAP, and an opposite one being represented by the ability of autophagy to degrade α-catenins, which act as negative vectors for YAP/TAZ activity. Consequently, when α-catenins levels are low, autophagy inhibits YAP/TAZ activity, and when α-catenins levels are high, autophagy upregulates YAP/TAZ activity. This is physiologically important as YAP/TAZ are oncogenes that regulate critical processes including cell growth, cell size and resistance to apoptosis.

## Results

**YAP/TAZ activity is inhibited by autophagy compromise in MCF10A cells**. We previously reported that the transcriptional regulators YAP/TAZ induce autophagy by altering the expression of actin cytoskeleton genes in various cell lines: MCF10A, HeLa, HaCaT cells, primary mammary epithelial cells (pMECs) and primary mouse embryonic fibroblasts (pMEFs)[21]. Here, we examined if an initial perturbation in the autophagy pathway would impact YAP/TAZ activity. Depletion of key genes governing autophagosome biogenesis by single ATG16L1 or double ATG7/ATG10 knockdowns in MCF10A cells shifted YAP/TAZ localisation from the nucleus (where they are transcriptionally active) to the cytoplasm (where they are inactive) (Fig. 1a, b), and

decreased TEAD luciferase activity (a reporter for YAP/TAZ activities, since YAP/TAZ activate TEAD transcription factors) (Fig. 1c and Supplementary Fig. 1). We confirmed these data in HEK293T cells, which are widely used to study Hippo signalling (Supplementary Figs. 2 and 3). Importantly, reduced YAP/TAZ nuclear activities were also seen in pMECs isolated from mice with a hypomorphic mutation in *Atg16L1*, which results in only modest impairment of autophagy[12,22] (Fig. 1d–f).

Consistent with the impaired actin stress-fibre formation known to result from YAP/TAZ inhibition[21], we observed reduced stress-fibre formation and myosin-II levels in autophagy-compromised cells (Fig. 1d, g and Supplementary Fig. 4). In addition, cell proliferation[23–26] (as assessed by BrdU positivity) and cell area[27,28], which are positively regulated by YAP/TAZ, were significantly reduced in autophagy-depleted MCF10A cells (Fig. 1h–j and Supplementary Fig. 5). Similar phenomena were observed for pMECs (Fig. 2a, b and Supplementary Fig. 6a), HEK293T (Supplementary Fig. 2d) and HeLa cells (as shown previously[21]). In 3D cell culture systems, one of the hallmarks of reduced YAP/TAZ activity is the formation of spherical cellular structures (acini)[29–31]. Indeed, in soft-matrix conditions (characterised by inactive, cytoplasmic YAP/TAZ[21,32]), control MCF10A cells form spherical acini structures, while, when grown in stiff extracellular collagen matrix, the cells start to lose their intimate contacts and spread to form arborescent, disorganised cellular branches. Interestingly, upon autophagy inhibition this stiff-matrix phenotype was reversed to soft-like cellular compact structures (Fig. 2c, d). Moreover, the diameter of the acini was diminished upon downregulation of autophagy genes (Fig. 2e–g).

We next investigated the effect of decreased Atg16L1 levels on the development of mammary acini. Mammary glands isolated from Atg16hyp (hypomorph) female mice at E16.5 gestation had reduced YAP/TAZ immunostaining intensity and smaller acini structures (area and interior perimeter), compared to their wild-type counterparts (Supplementary Fig. 6b), confirming the previous data achieved while culturing pMECs isolated from those tissues (Fig. 1d–g, Fig. 2a,b, d–g).

Similar 3D culture phenotypes associated with impaired YAP/TAZ activity were achieved upon either lysosome inhibition with bafilomycin A1 (BafA1), which blocks autophagosome degradation (Supplementary Fig. 7a), or with chemical suppression of autophagosome biogenesis using a VPS34 selective inhibitor, VPS34-IN1[33] (Supplementary Fig. 7b). Thus, autophagy inhibition impairs YAP/TAZ nuclear localisation and activities, leading to cellular phenotypes consistent with suppression of these transcriptional regulators.

**YAP/TAZ activity is upregulated by autophagy activation in MCF10A cells**. To examine the consequences on YAP/TAZ when autophagy is upregulated, we either cultured confluent MCF10A cells in Earle's Balanced Salt Solution (EBSS) (to mimic starvation-induced autophagy) or exposed them to well characterised autophagy activators: Tat-Beclin1, Trehalose or SMER28. All these conditions induced nuclear translocation of YAP/TAZ, the formation of stress fibres and increased the cell area in 2D cultures (Fig. 3a–i, Fig. 4a–f). Confluent MCF10A cells, with lower initial YAP/TAZ nuclear fractions[21], were used in these experiments in order to emphasise the YAP/TAZ activation mediated by autophagy induction. In 3D soft-matrix conditions, these autophagy enhancers promoted the formation of branched disorganised structures rather than spherical organoids (Fig. 4g–j).

**Direction of the YAP/TAZ response to autophagy perturbation is inversed in HepG2 cells**. Intriguingly, our data outlined above

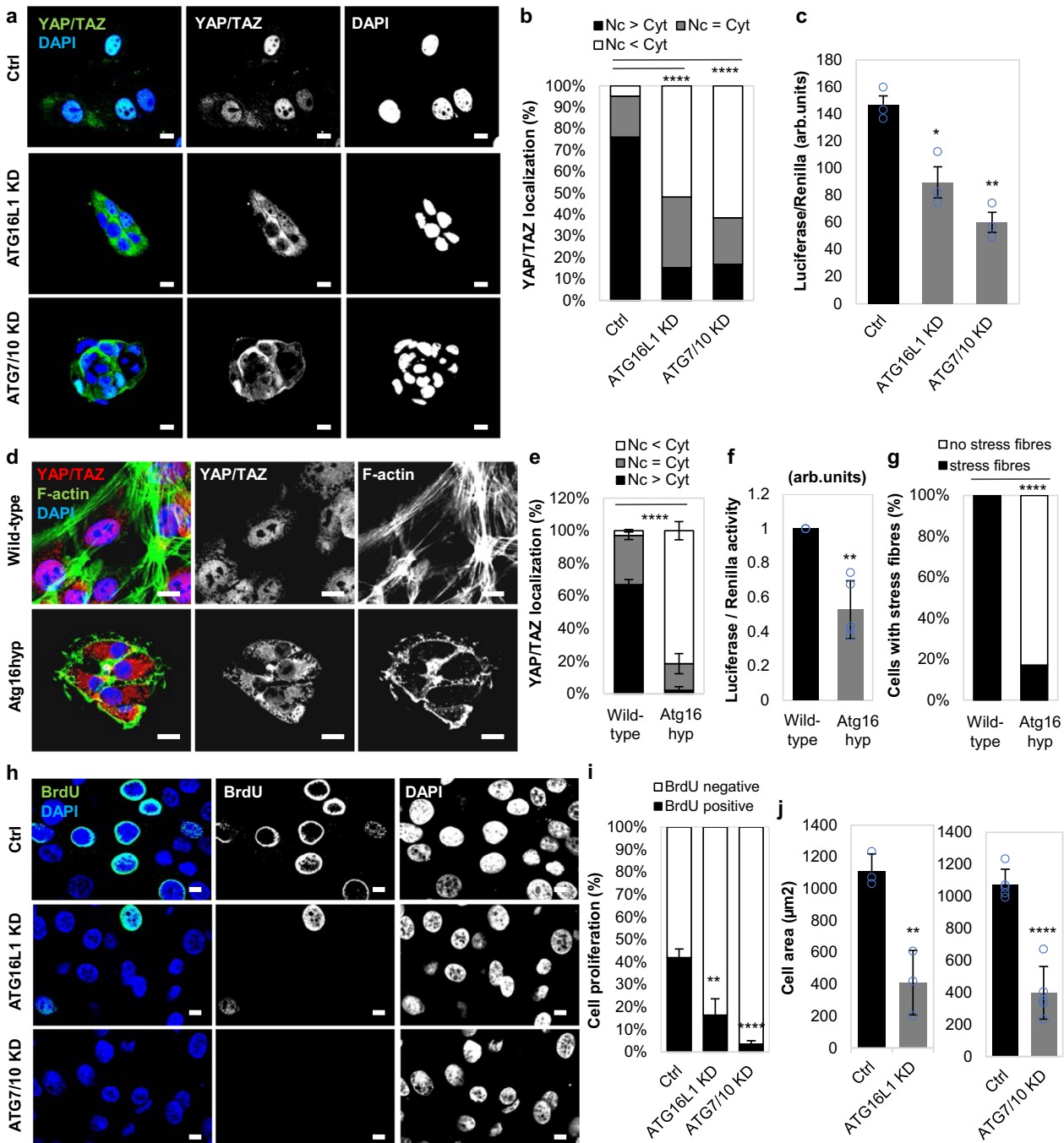

**Fig. 1 YAP/TAZ activity is inhibited in autophagy-deficient mammary epithelial cells. a** Representative confocal images of YAP/TAZ endogenous immunostaining in MCF10A cells exposed to control, ATG16L1 or ATG7/10 siRNAs. DAPI = nucleus. Scale bars are 10 μm. The experiment was repeated twice with similar results. **b** YAP/TAZ localisation (nuclear—Nc or cytoplasmic—Cyt) in MCF10A cells exposed to control, ATG16L1 or ATG7/10 siRNAs (*n* = 126 (Ctrl), 170 (ATG16L1 KD), 166(ATG7/10 KD) cells; ****P < 0.0001; chi-squared test). The experiment was repeated twice with similar results. **c** Luciferase assay for YAP/TAZ activity in MCF10A cells exposed to control, ATG16L1 or ATG7/10 siRNAs. Bars represent the mean ± s.d. (*n* = 3 independent experiments; **P < 0.01, *P < 0.05; two-tailed one sample *t*-test). **d** Representative confocal images of endogenous YAP/TAZ and F-actin (Phalloidin) in primary mammary epithelial cells (pMECs) isolated from wild-type and Atg16L1 hypomorph (Atg16hyp) mice. Scale bars are 10 μm. The experiment was repeated twice with similar results. **e** YAP/TAZ localisation in wild-type (*n* = 3 mice) and Atg16hyp (*n* = 4 mice) pMECs. Bars represent the mean ± s.d. (****P < 0.0001; two-way ANOVA). **f** Luciferase assay for YAP/TAZ activity in wild-type and Atg16hyp pMECs. Bars represent the mean ± s.d. (*n* = 3 mice; **P < 0.01; two-tailed one sample *t*-test). **g** Percentage of cells with F-actin stress fibres in wild-type and Atg16hyp pMECs (*n* = 98 (wild-type) and 292 (Atg16hyp) cells; ****P < 0.0001; chi-squared test). The experiment was repeated twice with similar results. **h** Representative images of BrdU immunostaining in MCF10A cells exposed to control, ATG16L1 or ATG7/10 siRNAs. Scale bars are 10 μm. The experiment was repeated twice with similar results. **i** The graphs show the percentages of BrdU-positive MCF10A cells, after exposure to control, ATG16L1 or ATG7/10 siRNAs: mean ± s.d. (*n* = 3 independent experiments; ****P < 0.0001, **P < 0.01; two-tailed *t*-test). **j** Size of autophagy inhibited cells: MCF10A cells exposed to control, ATG16L1 (*n* = 3 independent experiments) or ATG7/10 (*n* = 5 independent experiments) siRNAs. Confocal images of each cell type were analysed to measure cell area using ZEN software. Bars represent the mean ± s.d. (****P < 0.0001, **P < 0.01; two-tailed *t*-test). Exact *P* values for asterisks: **c** (from left to right) 0.0132, 0.0010; **f** 0.0033; **i** 0.0058; and **j** 0.0061.

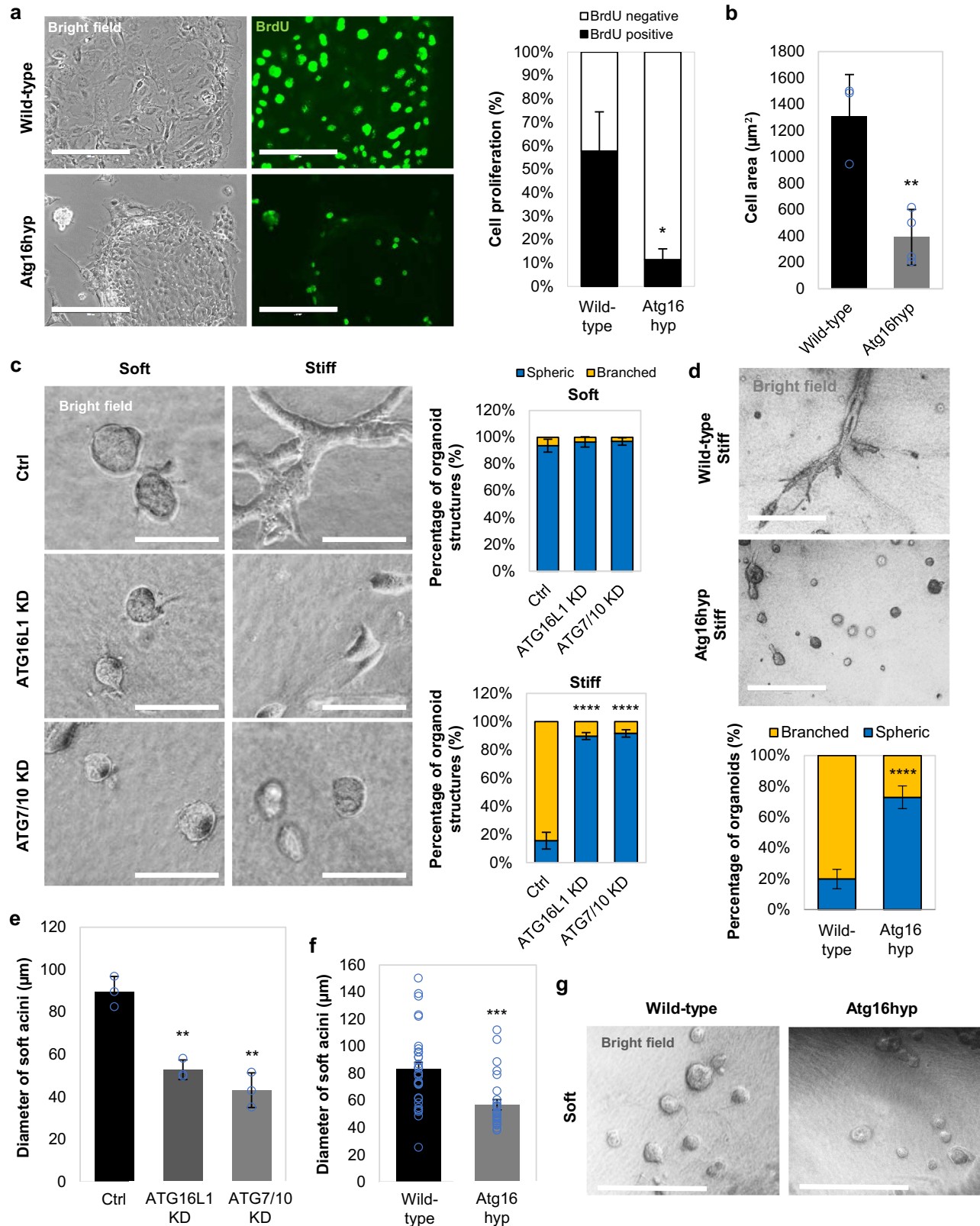

contrast with previous published literature claiming that autophagy inhibition activates YAP/TAZ and promotes their nuclear localisation. This was attributed to YAP being a direct autophagy substrate[18,19]. Accordingly, we investigated the effect of autophagy inhibition in HepG2 cells, as one of these two studies focused on liver systems[18]. Consistent with the published data,

downregulation of key autophagy genes in HepG2 cells promoted YAP/TAZ activity and nuclear localisation (Supplementary Fig. 8). Therefore, we hypothesised that the cells we studied (MCF10A, HEK293, HeLa cells and pMECs) have means to counter autophagic degradation of YAP[18,19], in order to increase YAP/TAZ activity in parallel with autophagic induction.

**Fig. 2 Formation of acini structures is perturbed in autophagy-deficient primary mammary epithelial cells. a** (left) Representative bright-field images and BrdU immunostaining in primary mammary epithelial cells (pMECs) isolated from wild-type and Atg16L1 hypomorph (Atg16hyp) mice. Scale bars are 200 μm. (right) Percentage of BrdU-positive pMECs isolated from wild-type ($n = 2$) and Atg16hyp ($n = 3$) mice. Bars represent the mean ± s.d. (*$P < 0.5$; two-way ANOVA). **b** Size of pMECs isolated from wild-type ($n = 3$) and Atg16hyp ($n = 4$) mice. Bars represent the mean ± s.d. (**$P < 0.01$; two-tailed $t$-test). **c** Representative confocal images of MCF10A cells plated on either soft or stiff ECM. MCF10A cells were previously exposed to control, ATG16L1 or ATG7/10 siRNAs. Scale bars are 200 μm. The bars represent the percentage of organoid/ spherical structures in soft ($n = 3$ independent experiments) or stiff ($n = 5$ independent experiments) ECM: mean ± s.d. (****$P < 0.0001$; two-tailed $t$-test). **d** Representative images of wild-type ($n = 3$ independent experiments) and Atg16hyp ($n = 5$ independent experiments) pMECs plated on stiff ECM. Scale bars are 400 μm. The bars represent the percentage of branched structures: mean ± s.d. (****$P < 0.0001$; two-tailed $t$-test). **e** Diameter of soft acini in MCF10A cells exposed to control, ATG16L1 or ATG7/10 siRNAs. Bars represent the mean ± s.d. ($n = 3$ independent experiments; **$P < 0.01$; two-tailed $t$-test). **f** Diameter of soft acini in wild-type or Atg16hyp pMECs. Bars represent the mean ± s.e.m. ($n = 33$ (wild-type), $n = 27$ (Atg16hyp) soft acini; ***$P < 0.001$; two-tailed $t$-test). The experiment was repeated with similar results. **g** Representative images of wild-type and Atg16hyp pMECs plated on soft ECM. Scale bars are 400 μm. The experiment was repeated twice with similar results. Exact $P$ values for asterisks: **a** 0.0163; **b** 0.0050; **e** (from left to right) 0.0017, 0.0018; and **f** 0.0001.

## Autophagy inhibition causes accumulation of α-catenins.

In order to identify plausible autophagy substrates responsible for YAP/TAZ inhibition, we performed SILAC experiments and identified the most abundant proteins that accumulated under prolonged BafA1 treatment (24 h). Among well-known autophagy substrates (NCOA4, TAX1BP1, NBR1, SQSTM/p62) we also identified CTNNA3, one of the three α-catenins (Fig. 5a and Supplementary Fig. 9a). α-Catenin is known to sequester and inhibit YAP/TAZ in the cytosol[34] and this may explain the why YAP/TAZ are inhibited in the cell lines we initially studied after autophagy inhibition. We confirmed the SILAC data by showing that endogenous α-catenins (CTNNA3 and CTNNA1) accumulated after double ATG7/ATG10 knockdowns (Fig. 5b, c and Supplementary Fig. 9b, c), BafA1 treatment (Supplementary Fig. 9d), or single ATG16L1 knockdown (Supplementary Fig. 9e, f) in MCF10A cells. Indeed, primary neurons (Fig. 5d) and pMECs isolated from Atg16hyp mice (Supplementary Fig. 9g) had higher levels of endogenous α-catenins, compared to their wild-type littermates. Autophagy inducers (EBSS, Tat-Beclin1) had the opposite effect of reducing endogenous levels of α-catenins (Fig. 5e–g). BafA1 enhanced the levels of transiently expressed mEm-CTNNA1 (Fig. 5h), while SMER28 and EBSS reduced the levels of mEm-CTNNA1 or Flag-CTNNA1 in dose- and time-dependent manners (Fig. 5i and Supplementary Fig. 9h, i).

## α-Catenins colocalise and directly interact with LC3.

We were struck that endogenous CTNNA3 and overexpressed mEm-CTNNA1 colocalised with endogenous LC3 (a marker for autophagosomes) (Fig. 6a, b), and this colocalization was even more obvious after Tat-beclin1 administration (Fig. 6c). Live-cell imaging showed enrichment of mRFP-LC3-positive structures (autophagosomes and autolysosomes) with mEm-CTNNA1 in MCF7 (Supplementary Fig. 10) and HeLa cells (Supplementary Fig. 11). The CTNNA1–LC3 interaction was further confirmed by in vitro binding assays (Supplementary Fig. 12) and by co-immunoprecipitation experiments between endogenous or overexpressed α-catenins and LC3 (Fig. 6d–f). In addition, BafA1 treatment enriched mEm-CTNNA1 accumulation in LC3-positive vesicles (Supplementary Fig. 13) and increased LC3–CTNNA1 (either endogenous or exogenous) interactions (Supplementary Fig. 14). These data suggest that CTNNA1 is indeed an autophagy substrate and accumulates in autophagosomes.

As autophagosomes ultimately fuse with lysosomes, we next confirmed a significant increase in colocalisation of CTNNA3 with the lysosomal marker, LAMP1 upon BafA1 treatment, which increases lysosomal pH and compromises its degradative capacity (400 nM, 4 h—Supplementary Fig. 15).

## α-Catenins interact with LC3 via LIR domains.

As we observed interactions between endogenous or overexpressed α-catenins and LC3 (Fig. 6d–f), we next searched for potential LC3-interacting regions (LIR) sequences in the structure of α-catenins and identified seven such putative motifs (Supplementary Fig. 16). We mutated four of them (where the amino acids face outwards—Supplementary Fig. 17) and identified highly conserved sites, CTNNA1(Y419A-V422A) and CTNNA1(L897A-F900A), where mutations led to much less LC3 being bound, compared to wild-type CTNNA1, using both GFP- and Flag-trap methods (Fig. 6e–g and Supplementary Fig. 14). Indeed, these two CTNNA1 mutants did not colocalise with LC3, showing a preferentially perinuclear localisation (Fig. 6h, i and Supplementary Fig. 13). As expected, the single CTNNA1(Y419A-V422A), CTNNA1(L897A-F900A) and double CTNNA1(Y419A-V422A-L897A-F900A) mutants (tagged with either mEmerald or Flag) did not accumulate upon BafA1 treatment (400 nM, 6 h) (Fig. 7a, b) and were not cleared as effectively as the wild-type by autophagy upregulation with SMER28 (20 μM) or EBSS for 6 h (Fig. 7c–e) in MCF10A cells. These data led us to conclude that CTNNA1 is an autophagy substrate that directly interacts with LC3 via putative LIR motifs.

## Wild-type α-catenins, and not LIR-defective mutants, are autophagy substrates.

As CTNNA1 interacts with YAP/TAZ, we considered if the CTNNA1-LC3 interaction could account for the decrease in YAP/TAZ activity after autophagy inhibition in the cells we studied. Overexpression of CTNNA1(Y419A-V422A), CTNNA1(L897A-F900A) and CTNNA1(Y419A-V422A-L897A-F900AA) mEmerald-tagged mutants reduced the nuclear localisation of YAP/TAZ, increasing its cytosolic fraction (Fig. 8a, b, Supplementary Figs. 18, 19) to levels similar to those achieved upon BafA1 treatment (Fig. 8c). These mutants, which have compromised LC3 binding and autophagic clearance, had more marked effects of reducing the YAP nuclear fraction than wild-type CTNNA1, suggesting that YAP/TAZ cytosolic retention in our autophagy-compromised cells may be due to reduced autophagic degradation of α-catenins. The cytosolic YAP retention by LIR-defective α-catenins was confirmed by increased cytosolic co-immunoprecipitation of GFP-YAP with Flag-tagged CTNNA1 mutants when compared to the wild-type form (Supplementary Fig. 20a). BafA1 treatment doubled the amount of YAP co-immunoprecipitated with wild-type CTNNA1 (Supplementary Fig. 20a), while EBSS caused the opposite effect (of decreasing the interaction)—Supplementary Fig. 20b. Conversely, LIR-defective α-catenins did not show a similar behaviour of increased interactions with YAP upon BafA1 treatment, or reduced interactions upon exposure to EBSS (Supplementary Fig. 20). These data suggest a LIR/autophagy-dependent mechanism for YAP-cytosolic sequestration by α-catenins. The LIR-defective CTNNA1 mutants also reduced the nuclear YAP/

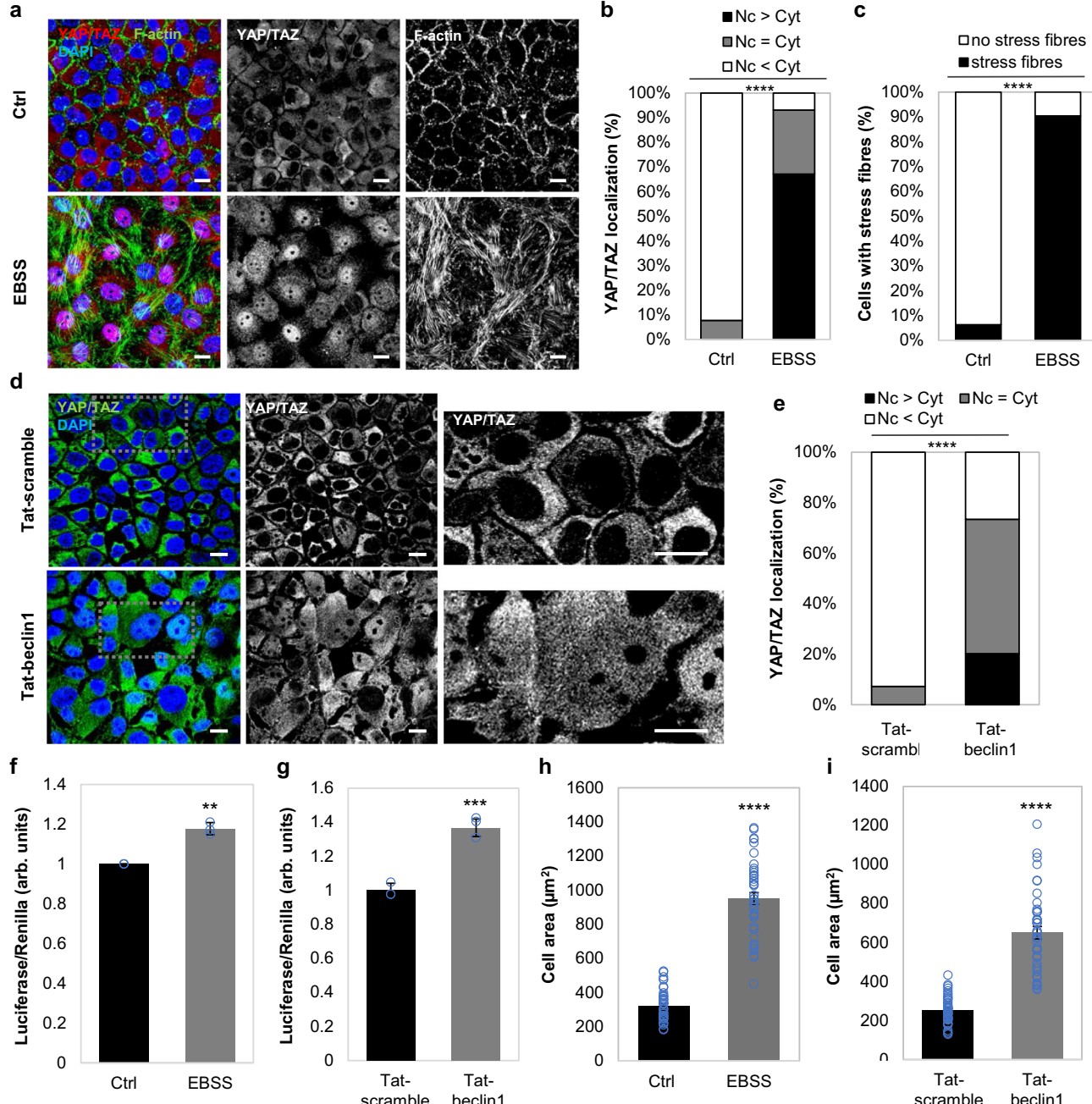

**Fig. 3 YAP/TAZ is activated upon autophagy induction (EBSS, Tat-beclin1) in MCF10A cells. a** Representative confocal images of YAP/TAZ and F-actin (phalloidin) immunostaining in MCF10A cells cultured at high confluency in EBSS for 6 h. Scale bars are 10 µm. The experiment was repeated twice with similar results. **b** YAP/TAZ localisation in MCF10A cells exposed to EBSS for 6 h ($n = 143$ cells (Ctrl), $n = 146$ cells (EBSS); ****$P < 0.0001$; chi-squared test). The experiment was repeated twice with similar results. **c** Percentage of MCF10A cells with stress fibres upon starvation (EBSS, 6 h) ($n = 143$ cells (Ctrl), $n = 146$ cells (EBSS); ****$P < 0.0001$; chi-squared test). The experiment was repeated twice with similar results. **d** Representative confocal images of YAP/TAZ immunostaining in MCF10A cells cultured at high confluency and exposed to either Tat-scramble (control) or Tat-beclin1 peptide (20 µM, 48 h). Scale bars are 10 µm. The experiment was repeated twice with similar results. **e** YAP/TAZ localisation in MCF10A cells exposed to either Tat-scramble (control) or Tat-beclin1 peptide (20 µM, 48 h) (****$P < 0.0001$; chi-squared test). The experiment was repeated with similar results. **f** TEAD luciferase activity in MCF10A cells starved with EBSS for 6 h. Bars represent the mean ± s.d. ($n = 3$ independent experiments; **$P < 0.01$; two-tailed one sample t-test). **g** TEAD luciferase activity in MCF10A cells treated with Tat-beclin1 peptide (20 µM, 48 h). Bars represent the mean ± s.d. of a representative experiment performed in triplicates (***$P < 0.001$; two-tailed t-test). **h** Size of MCF10A cells starved in EBSS. Bars represent the mean ± s.e.m. ($n = 42$ cells, ****$P < 0.0001$; two-tailed t-test). **i** Size of MCF10A cells treated with Tat-beclin1 peptide (20 µM, 48 h). Bars represent the mean ± s.e.m. ($n = 45$ cells; ****$P < 0.0001$; two-tailed t-test). Exact $P$ values for asterisks: **f** 0.0096; and **g** 0.0006.

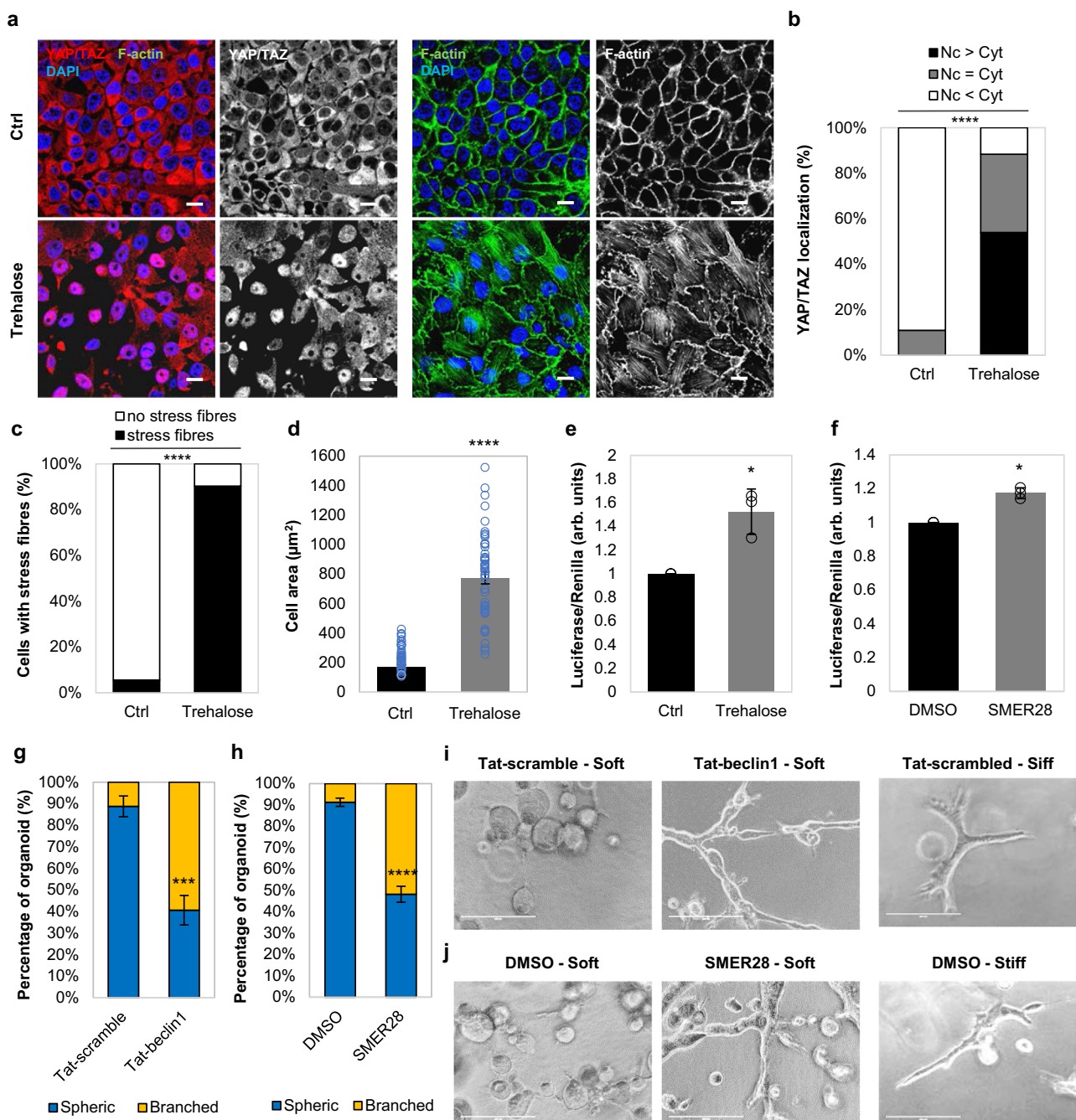

**Fig. 4 YAP/TAZ is activated upon autophagy induction (Trehalose, SMER28) in MCF10A cells. a** Representative confocal images of YAP/TAZ and F-actin (phalloidin) immunostaining in MCF10A cells cultured at high confluency and exposed to trehalose (100 mM for 24 h). Scale bars are 10 μm. The experiment was repeated at least once with similar results. **b** YAP/TAZ localisation in MCF10A cells exposed to trehalose (100 mM for 24 h) with at least 100 cells analysed per condition (****$P < 0.0001$; chi-squared test). The experiment was repeated with similar results. **c** Percentage of MCF10A cells with stress fibres exposed to trehalose (100 mM for 24 h) ($n = 100$ cells; ****$P < 0.0001$; chi-squared test). The experiment was repeated with similar results. **d** Cell area of MCF10A cells exposed to trehalose (100 mM for 24 h). Bars represent the mean ± s.e.m. ($n = 50$ cells; ****$P < 0.0001$; two-tailed $t$-test). **e** TEAD luciferase activity of MCF10A cells treated with trehalose (100 mM for 24 h). Bars represent the mean ± s.d. ($n = 3$ independent experiments; *$P < 0.05$; two-tailed one sample $t$-test). **f** TEAD luciferase activity of MCF10A cells treated with SMER28 (50 μM, 6 h). Bars represent the mean ± s.d. ($n = 3$ independent experiments; *$P < 0.05$; two-tailed one sample $t$-test). **g** Quantification of branched and spherical cellular structures of MCF10A cells exposed to either Tat-scrambled (control) or Tat-beclin1 peptide (20 μM, 48 h), as in (**i**). Bars represent the mean ± s.e.m. ($n = 3$ independent experiments; ***$P < 0.001$; two-tailed $t$-test). **h** Quantification of branched and spherical cellular structures of MCF10A cells exposed to either DMSO (control) or SMER28 (50 μM) as in (**j**). Bars represent the mean ± s.e.m. ($n = 3$ independent experiments; ****$P < 0.0001$; two-tailed $t$-test). **i** Representative bright-field images of MCF10A cells exposed to Tat-beclin1 peptide (20 μM, 48 h) and cultured in soft 3D extracellular matrix. Tat-scrambled was used to treat the control cells. Scale bars are 200 μm. The experiment was repeated twice with similar results. **j** Representative bright-field images of MCF10A cells exposed to SMER28 (50 μM) and cultured in soft 3D extracellular matrix. DMSO was used to treat the control cells. Scale bars are 200 μm. The experiment was repeated twice with similar results. Exact $P$ values for asterisks: **e** 0.0421; **f** 0.0113; and **g** 0.0006.

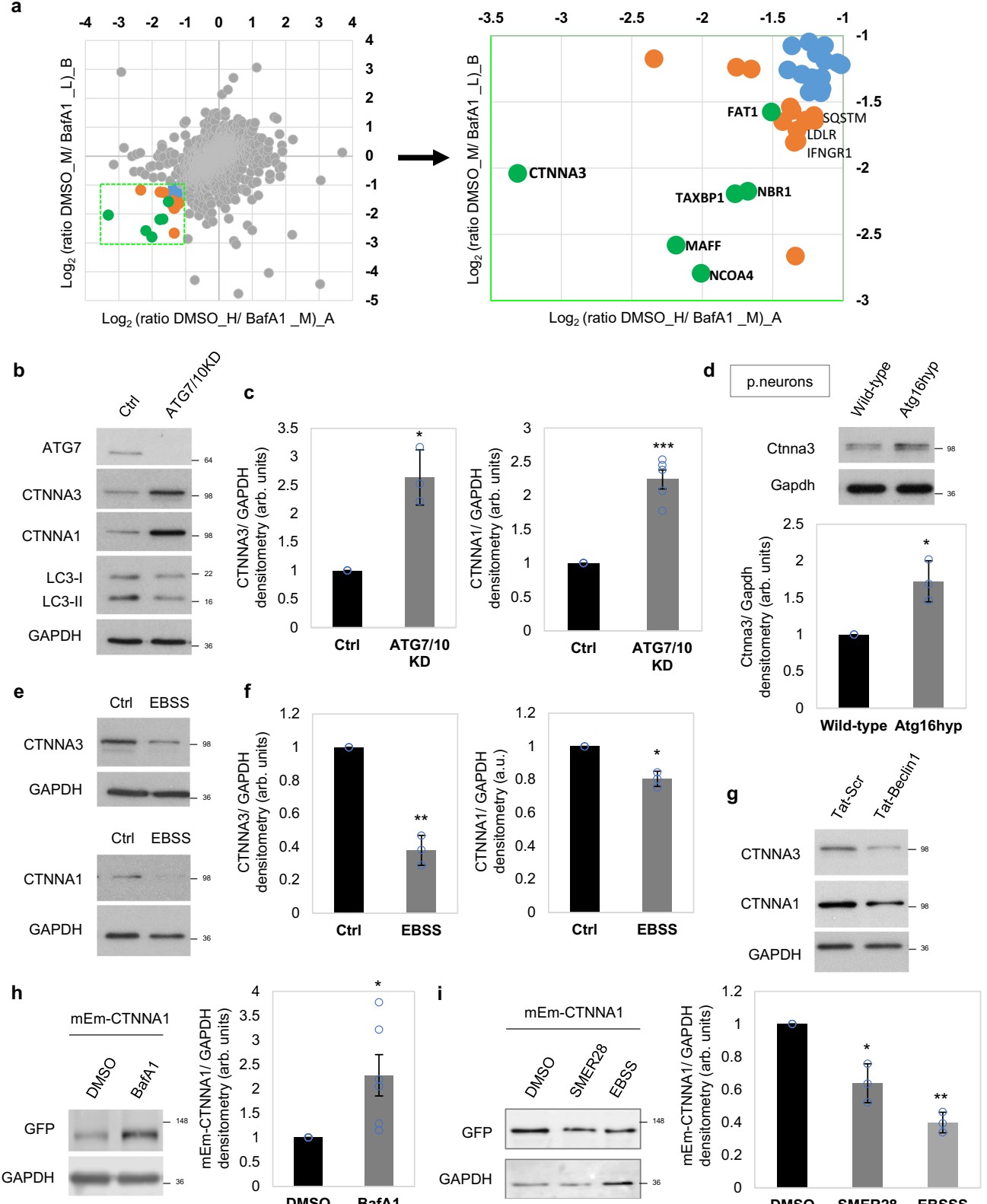

TAZ localisation more markedly than wild-type CTNNA1 when expressed in CTNNA1-depleted cells and CTNNA1 knockdown increased the YAP/TAZ nuclear localisation, which was reduced upon CTNNA1 (wild-type) re-expression (Fig. 8d). TEAD luciferase activity (Fig. 8e), cell proliferation (Fig. 8f, g, Supplementary Fig. 19c), and cell area (Fig. 8h), additional correlates of

YAP/TAZ activities, were all decreased in MCF10A cells expressing the LIR-defective CTNNA1 mutants to a greater extent than the wild-type protein. To further confirm the role of α-catenins (CTNNA1, CTNNA3) as effectors for the YAP/TAZ phenotype caused by autophagy perturbation, MCF10A cells were initially exposed to ATG7/10 siRNAs for 48 h in order to reach an

**Fig. 5 α-Catenins are autophagy substrates. a** Scatterplot of all proteins uniquely identified in both SILAC experiments (L light, M medium, H heavy amino acids). Fold change (x-axis and y-axis for experiment A and B, respectively) are shown as log2. Scatterplot of the proteins upregulated ≥ 2 fold under bafilomycin A1 (BafA1) treatment is shown on the right. **b** Representative α-catenins (CTNNA3 and CTNNA1) immunoblotting in double ATG7/10 knockdown MCF10A cells. The experiment was repeated at least twice with similar results. **c** CTNNA3/GAPDH (left panel) and CTNNA1/GAPDH (right panel) densitometry in MCF10A cells exposed to either control or ATG7/ATG10 siRNAs. The graphs show the mean ± s.d. (CTNNA3/ GAPDH, $n = 3$ independent experiments) and mean ± s.e.m. (CTNNA1/ GAPDH, $n = 5$ independent experiments; ***$P < 0.001$, *$P < 0.05$; two-tailed one sample $t$-test). **d** Representative immunoblots of Ctnna3 levels in wild-type and Atg16L1 hypomorph (Atg16 hyp) primary cortical neurons. The graph shows the Ctnna3/ Gapdh levels: mean ± s.d. ($n = 3$ independent experiments; *$P < 0.5$; two-tailed one sample $t$-test). **e** Representative α-catenins (CTNNA3 and CTNNA1) immunoblots in MCF10A cells under starvation with EBSS (6 h). The experiment was repeated at least twice with similar results. **f** CTNNA3/GAPDH (left panel) and CTNNA1/GAPDH (right panel) densitometry in MCF10A cells starved in EBSS for 6 h. The graphs show the mean ± s.d. ($n = 3$ independent experiments; **$P < 0.01$, *$P < 0.05$; two-tailed one sample $t$-test). **g** Representative α-catenins (CTNNA3 and CTNNA1) immunoblots in MCF10A cells treated with the autophagy inducer Tat-Beclin1 peptide (20 μM for 48 h). Tat-scrambled peptide was used as control. The experiment was repeated at least twice with similar results. **h** Representative immunoblot of mEm-CTNNA1 in MCF10A cells exposed to BafA1 (200 nM, 16 h). The MCF10A cells were transiently transfected with mEm-CTNNA1. The graph shows the mEm-CTNNA1/GAPDH levels: mean ± s.d. ($n = 6$ independent experiments; *$P < 0.05$; two-tailed one sample $t$-test). **i** Representative immunoblot of mEm-CTNNA1 in MCF10A cells exposed to autophagy inducers: SMER28 (20 μM) and EBSS for 6 h. The graph shows the mEm-CTNNA1/GAPDH levels: mean ± s.d. ($n = 3$ independent experiments; **$P < 0.01$, *$P < 0.05$; two-tailed one sample $t$-test). Exact $P$ values for asterisks: **c** (from left to right) 0.0280, 0.0009; **d** 0.0461; **f** (from left to right) 0.0069, 0.0181; **h** 0.0301; and **i** (from left to right) 0.0340, 0.0036.

autophagy-deficient status, and only after knockdowns, α-catenins were depleted using CTNNA1/3 siRNAs for other 48 h. While ATG7/10 knockdown caused accumulation of α-catenins and subsequent YAP-cytosolic sequestration (Fig. 9a–b) with consequent changes in the actin cytoskeleton morphology (Fig. 9c) and cell area (Fig. 9d) in MCF10A cells, the subsequent knockdown of α-catenins in the autophagy-compromised cells was, indeed, able to partially rescue those previously perturbed YAP/TAZ-related phenotypes (Fig. 9a–d and Supplementary Fig. 21). Thus, in cells like MCF10A, autophagy positively regulates YAP/TAZ activity by enhancing degradation of α-catenins, which otherwise sequester YAP/TAZ in the cytoplasm.

**α-Catenins levels determine differential autophagy effects on YAP/TAZ activity**. The inhibition of YAP/TAZ upon autophagy suppression was only seen in certain cell types: MCF10A, HEK293T, HeLa and pMECs (Figs. 1–4 and Supplementary Figs. 1–7). The opposite YAP/TAZ response is seen in other cells, including HepG2 (Supplementary Fig. 8), THLE2 and A549 cells[18,19]. As we described previously that YAP/TAZ positively regulate autophagosome biogenesis[21] and our current data suggest that α-catenins (known inhibitors of YAP/TAZ) are autophagy substrates, we considered whether a YAP/TAZ-autophagy-YAP/TAZ feedback loop operates in cells, with starting the levels of α-catenins explaining the output differences among various cellular systems. As autophagy is a dynamic process and we are postulating a feedback loop, we considered the possibility that one might see different effects on YAP/TAZ activity at various times after initiating the autophagy perturbation. To explore this possibility, we created a numerical mathematical model based on three differential equations to measure the dynamics of autophagy levels, YAP/TAZ activity and levels of α-catenins, when autophagy is primarily impaired. This model also took into account the effect of the previously described positive regulation of autophagy by YAP/TAZ (YAP/TAZ inhibition lowers autophagy levels, at both autophagosome biogenesis[21] and autophagosome-lysosome fusion[31] steps) and the direct effect of autophagy on degrading YAP[18]. Among the parameters we varied, the initial values of intracellular α-catenins (α-cat) and the strength of the feedback loop (the variable which controls the YAP/TAZ influence over autophagy, $v_Y$) had the highest impact on the behaviour of the autophagy-YAP/TAZ loop (Fig. 9e and Supplementary Fig. 22a). However, the initial intracellular values of α-catenins had the strongest influence over the YAP/TAZ

activity upon autophagy depletion in our mathematical model: higher initial α-cat values (>0.5 in Fig. 9e, corresponding to cells with basal α-catenins levels of at least 50 % of those found in MCF10A cells) promoted a reduction in YAP/TAZ activity, while small initial α-cat values (<0.3 in Fig. 9e, corresponding to cell lines with lower initial CTNNA1 levels compared to MCF10A cells by at least 70 %) promoted the opposite effect (of increased YAP/TAZ activity) when autophagy was primarily compromised.

To support these numerical findings, we next assessed the CTNNA1 protein levels in various cell lines. Indeed, cell lines like HEK293T, HeLa, pMECs, which behave similarly to MCF10A cells in relation to the autophagy-YAP/TAZ loop (cells where autophagy compromise inhibits YAP/TAZ activity and autophagy induction promotes YAP/TAZ activity) have higher CTNNA1 levels (the relative ratio of CTNNA1 basal levels > 0.5, in Fig. 9f,g). On the other hand, cell lines that show decreased YAP/TAZ signalling after autophagy induction (HepG2, THLE2, A549)[18,19] have lower CTNNA1 levels (the relative ratio of CTNNA1 basal levels being around 0.3 or less—Fig. 9f, g). Our modelling also predicts that cell lines characterised by lower α-cat will accumulate less α-catenins upon autophagy inhibition (Supplementary Fig. 22b). To corroborate this effect, ATG7 and ATG10 were depleted in various cell lines. Indeed, HepG2 (Fig. 10a) and Huh7 cells (Supplementary Fig. 23a), characterised by low α-cat values, showed small CTNNA1 increases of only 12% and 15%, respectively, when compared to MCF10A (125% increase) or HEK293T (85% increase) cells (Fig. 10a). The model may also predict a time-dependent effect on YAP/TAZ activity when the initial YAP activity (Supplementary Figs. 24 and 25) or autophagy (Supplementary Figs. 26 and 27) levels are varied. Specifically, higher initial YAP values (Y_init > 1) are predicted to cause slightly smaller relative increases in YAP activity (YAP/Y_init – bottom graph in Supplementary Fig. 24) when compared to lower initial YAP conditions (Y_init < 1) at initial time points, but only in the cells with low initial α-cat values (<0.3, Supplementary Figs. 24 and 25). For the cases characterised by high basal α-cat values (>0.5), the initial levels of YAP activity are not predicted to have major influences on the system outputs (Supplementary Figs. 24 and 25). When initial autophagy levels are varied (with Y_init = 1), the numerical model estimates that YAP/TAZ activity changes faster upon autophagy inhibition in systems characterised by basal lower autophagy (A_init < 1), compared to higher initial autophagy conditions (A_init > 1), and this effect is amplified when initial α-cat values are increased (Supplementary Figs. 26

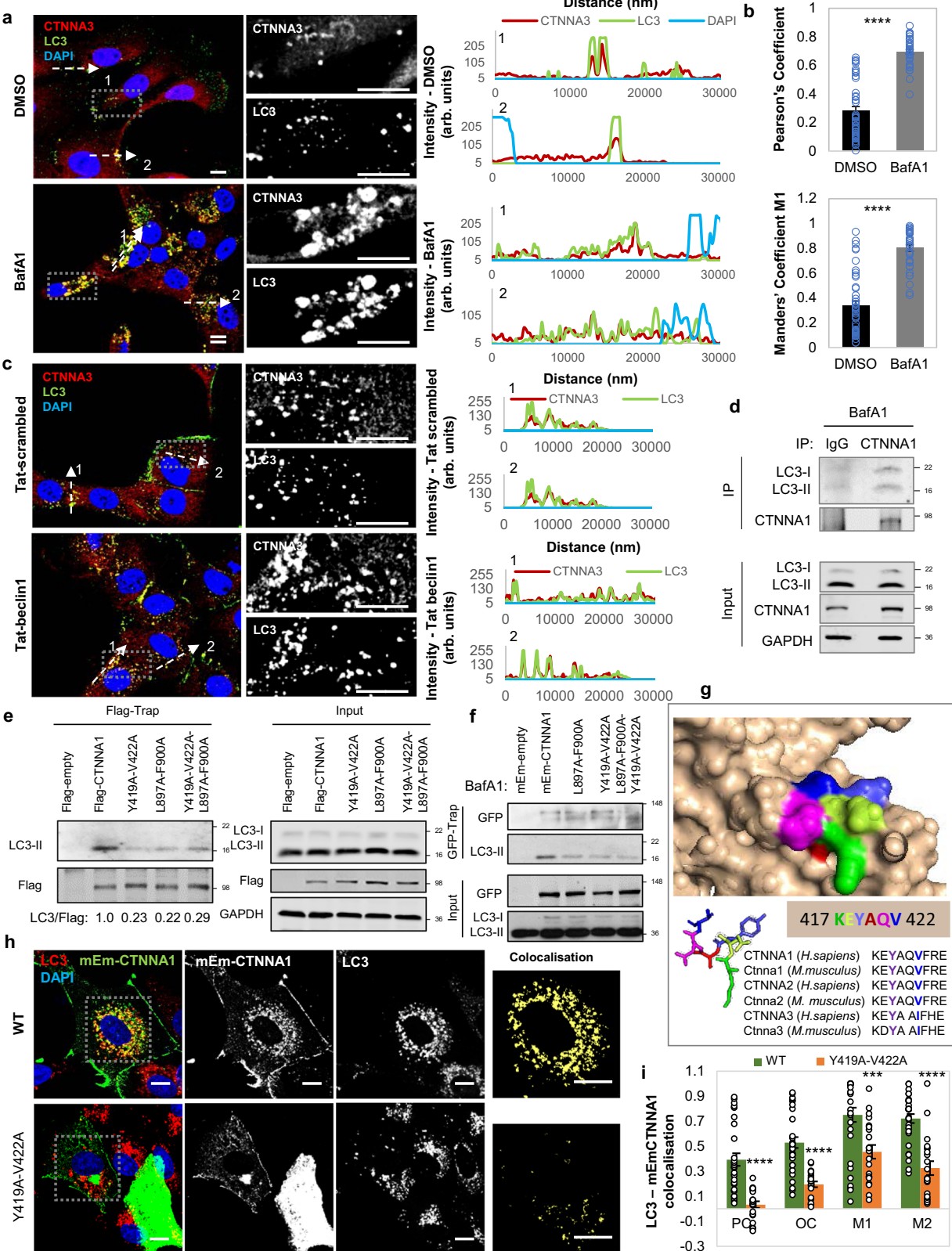

and 27). Thus, we may conclude from our numerical simulations that the initial levels of YAP activity appear to predominantly impact the magnitude of the output (YAP/TAZ activity), rather than influencing its direction (of increasing or decreasing) or its time period, while the initial autophagy levels mainly impact on the time-span of the output-effect.

Our modelling posits that starting levels of α-catenins determine the direction of the YAP/TAZ signalling response to autophagy perturbation (Fig. 9g, see also the extended explanation of the mathematical model in Supplementary Fig. 28), and can explain the divergent effects of autophagy perturbation on YAP/TAZ signalling in MCF10A cells (high starting levels of α-

**Fig. 6 α-Catenins interact with LC3 via two LIR regions. a** Representative images of endogenous CTNNA3–LC3 colocalization in MCF10A cells treated with BafA1. The control cells were treated with DMSO. Scale bars are 10 μm. The graphs on the right show the image intensities for all channels (CTNNA3, LC3 and DAPI). Scale bars are 10 μm. The experiment was repeated twice with similar results. **b** Quantification of colocalization for the experiment in (**a**). The Pearson's and Mander's (fraction of CTNNA3 overlapping LC3) coefficients were quantified for $n = 45$ cells (DMSO) and $n = 38$ cells (BafA1). Bars represent the mean ± s.e.m. (****$P < 0.0001$; two-tailed $t$-test). **c** Representative confocal images of endogenous CTNNA3–LC3 colocalisation in MCF10A cells exposed to either Tat-scrambled or Tat-beclin1 peptide (20 μM, 48 h). Scale bars are 10 μm. The graph on the right shows the fluorescent intensity for each channel (red-CTNNA3, green-LC3 and blue-DAPI) along the indicated distances. **d** Co-immunoprecipitation of endogenous LC3 with CTNNA1 in BafA1-treated MCF10A cells. The experiment was repeated three times with similar results. **e** Co-immunoprecipitation of endogenous LC3 with wild-type and Flag-CTNNA1 mutants in BafA1-treated MCF10A cells. MCF10A cells were initially transfected with empty Flag (control vector), wild-type Flag-CTNNA1 or the indicated Flag-CTNNA1 mutants. The Flag-tagged proteins were pulled down using the Flag-Trap technology. The experiment was repeated three times with similar results. **f** Co-immunoprecipitation of endogenous LC3 with wild-type and mEm-CTNNA1 mutants. MCF10A cells were transfected with empty mEm (control vector), wild-type mEm-CTNNA1 or the indicated mEm-CTNNA1 mutants. The mEm-tagged proteins were pulled down using the GFP-Trap technology. The experiment was repeated three times with similar results. **g** 3D structure of the LIR region (417 KE**Y**AQ**V** 422) and its conservation among various species of α-catenins. **h** Representative images of mEm-CTNNA1—endogenous LC3 colocalization in MCF10A cells transiently transfected with either wild-type or Y419A-V422A mEm-CTNNA1. Scale bars are 10 μm. **i** The Pearson's (PC), Overlap (OC) and Mander's (M1—fraction of LC3 overlapping mEm-CTNNA1, and M2—fraction of mEm-CTNNA1 overlapping LC3) coefficients for MCF10A cells treated as in (**h**). Thirty-one wild-type cells and 21 Y419A-V422A mutant cells were quantified over three independent experiments. Bars represent the mean ± s.e.m. (****$P < 0.0001$, ***$P < 0.001$; two-tailed $t$-test). Exact $P$ values for asterisks: **i** 0.0009.

catenins) versus HepG2 cells (low starting levels of α-catenins). To further test the central role of α-catenin levels in this model, we have engineered MCF10A cells to have low levels of α-catenins (using siRNA knockdowns) and HepG2 cells to have high levels of α-catenins by overexpression (Fig. 10 and Supplementary Figs. 29, 30). In MCF10A cells previously depleted of α-catenins by siRNA knockdowns for 48 h, untreated cells had higher YAP/TAZ activity than those exposed to autophagy stimuli with EBSS for 6 h (the experimental YAP/TAZ activity data are shown in Fig. 10b, while the corresponding modelling values are depicted in Supplementary Fig. 29a–c) or Trehalose 100 mM for 24 h (the experimental YAP/TAZ activity data are shown in Fig. 10c, while the corresponding modelling values are depicted in Supplementary Fig. 29d; the input YAP activity values in our modelling system were corresponded to those achieved experimentally by CTNNA1/3 knockdown under basal conditions). Conversely, a short-time exposure to ATG7/10 siRNAs (2 days) in MCF10A cells previously depleted of α-catenins for 48 h, increased TEAD-YAP luciferase activity (the experimental YAP/TAZ activity data are shown in Fig. 10d, while the corresponding modelling values are depicted by the orange vertical bar on the left in Supplementary Fig. 30a) similar to cells characterised by initial low levels of α-catenins, such as HepG2. Consistent with our model, in HepG2 cells overexpressing the mEm-CTNNA1 construct, EBSS treatment for 6 h enhanced the YAP/TAZ activity in the HepG2 cells (the experimental YAP/TAZ activity data are shown in Fig. 10e, while the corresponding modelling values are depicted in Supplementary Fig. 29e). Thus, the HepG2 cells overexpressing wild-type CTNNA1 phenocopied what we had seen in MCF10A cells (Fig. 3f). Overall, these experimental results suggest that initial levels of α-catenins determine the dynamics of YAP/TAZ response to various autophagy perturbations.

Since the autophagy perturbation is a dynamic process that spans over a certain time period, one should also consider the time dependency of autophagy perturbations on YAP/TAZ activity, detailed in Supplementary Figs. 28 and 30. It is interesting to note that our modelling predicts that in low initial α-catenin conditions, YAP/TAZ activity will increase as autophagy is inhibited at early time points (YAP increases while autophagy decreases—Supplementary Fig. 30b, $R$ correlation coefficient $< -0.5$). This effects is driven by the ability of autophagy to degrade YAP/TAZ[18,19] dominating the negative effect of impact of α-catenin accumulation on YAP/TAZ when α-catenin levels are low. However, when autophagy is inhibited then α-catenin also starts to increase and after

a certain period of time α-catenin levels will reach a threshold when α-catenin sequestration of YAP/TAZ dominates the autophagy-mediated degradation of YAP/TAZ. From this timepoint onwards, YAP decreases as autophagy inhibition continues—Supplementary Fig. 30b, c $R > 0.5$. In high initial α-catenin conditions, YAP/TAZ activity is predicted to positively correlate with autophagy levels at any relevant experimental time of the applied perturbation (Supplementary Figs. 24 and 26).

## Discussion

Collectively, our data suggest, focusing on the YAP/TAZ signalling output, that autophagy may impact on cell identity, causing completely opposite effects in cells with different starting points (different α-catenin levels). This is physiologically relevant, as the principles underlying the described mechanism and proposed mathematical model in this study may extend to other signalling pathways, which may respond to autophagy perturbations with contrasting reactions in distinct cells, tissues, organs or even individuals. This type of biology may explain many apparent discrepancies in the literature in signalling responses to different stimuli. These data further reveal how feedback loops enable the post-transcriptional determination of cell identity. Our results, both experimental and obtained by mathematical modelling, raise important points that need to be carefully considered for biological replication purposes: output differences in signalling pathways, here described for YAP/TAZ, are tightly conditioned by the initial experimental set up, the order and time-span of cellular perturbations.

These findings are directly relevant in explaining human pathology and may help define therapeutic strategies based on autophagy modulation—for example, autophagy inhibition may differentially affect oncogenic pathways in different directions in different cells types. Indeed, our data suggest that autophagy inhibition impairs YAP/TAZ activity and proliferation of primary mammalian epithelial cells (Fig. 1d–g, i), while autophagy compromise in hepatocytes induces YAP activity and increases lives size, cell proliferation and carcinogenesis[17,18].

## Methods

**Antibodies**. The following primary antibodies used for western blot (WB) and immunofluorescence (IF): rabbit polyclonal anti-ATG7 (ab52472, WB 1:1000, IF 1:200 dilution), rabbit monoclonal anti-ATG10 (ab124711, WB 1:1000), mouse monoclonal anti-BrdU (ab8152, IF 1:200), rat monoclonal anti-BrdU (ab6326, IF 1:100), mouse monoclonal anti-GAPDH (ab8245, WB 1:5000), goat polyclonal anti-GFP (ab5450, IF 1:200), rabbit polyclonal anti-GFP (ab6556, WB 1:1000), rabbit monoclonal anti-LC3 (ab192890, WB 1:1000), rabbit monoclonal anti-

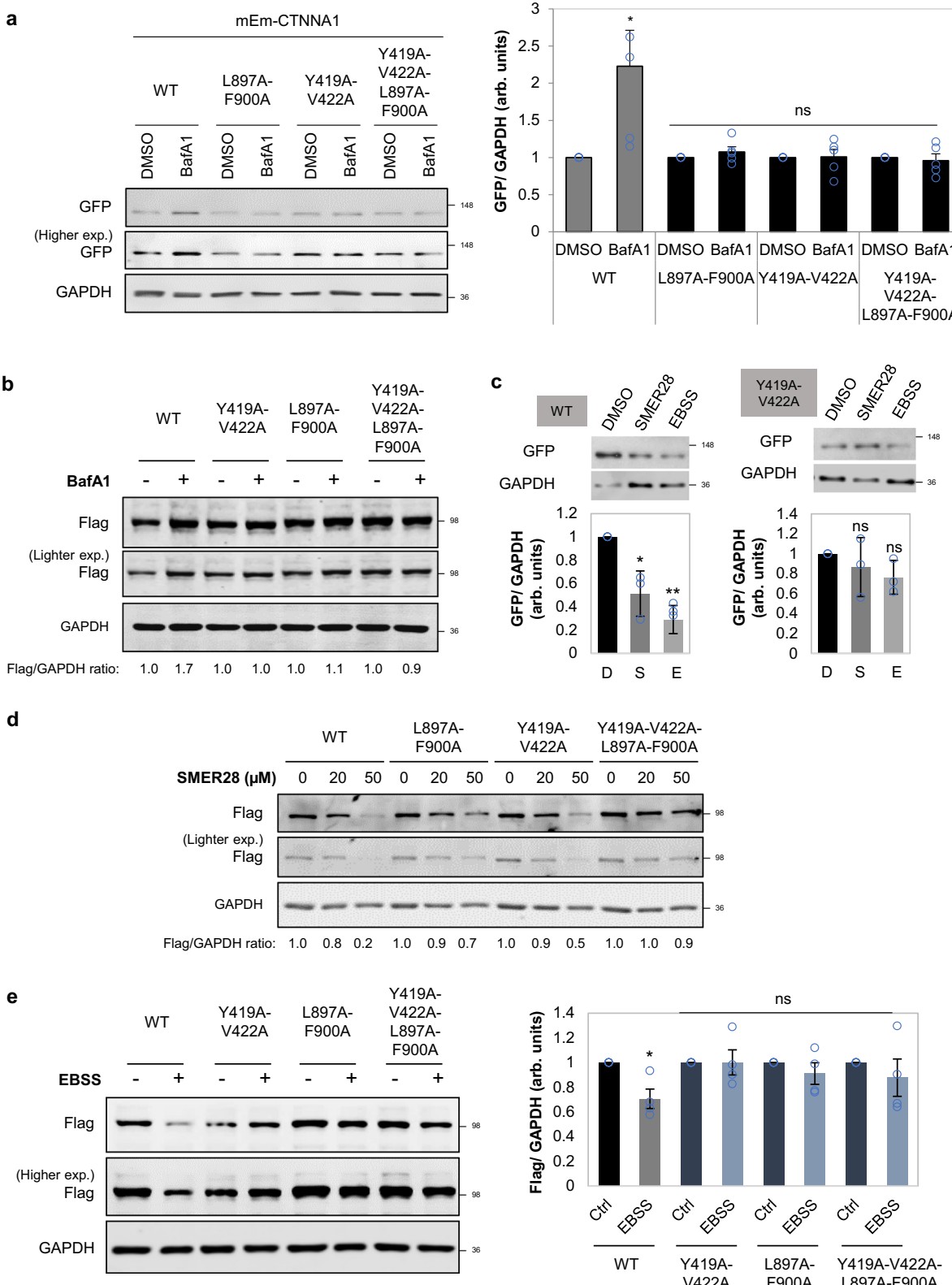

CTNNA3 (ab184916, WB 1:1000) from Abcam. Mouse monoclonal anti-LC3 (clone 5F10, Nanotools, IF 1:200), mouse monoclonal anti-YAP/TAZ (63.7) (sc-101199, Santa Cruz Biotechnology, IF 1:100), goat polyclonal anti-LaminB (M-20) (sc-6217, Santa Cruz Biotechnology, WB 1:100), mouse monoclonal anti-LAMP1 (ab25630, IF: 1:100), rabbit monoclonal anti-S6 Ribosomal Protein (5G10) (no. 2217, Cell Signalling, WB 1:1000), rabbit monoclonal anti-α-E-catenin (anti-CTNNA1) (no. 3236, Cell Signalling, WB 1:1000), rabbit monoclonal anti-MLC2

(no. 8505, Cell signalling, WB 1:1000), normal rabbit IgG (no. 2729S, Cell Signalling), rabbit polyclonal anti-ATG16L1 (no. 8089, Cell Signalling, IF 1:200), rabbit anti-ATG16L1 (PM040, MBL, WB 1:1000), mouse anti-Flag M2 (F3165, Sigma-Aldrich, WB: 1:2000).

The secondary antibodies used for IF were conjugated to Alexa Fluor 488, 568, 594 or 647 (Invitrogen). The horseradish peroxidase-conjugated secondary antibodies used for western blotting were as follows: anti-mouse (NA931V, GE

**Fig. 7 LIR-defective CTNNA1 mutants are not autophagy substrates. a** Representative immunoblots of MCF10A cells transiently expressing wild type, L897A-F900A, Y419A-V422A or Y419A-V422A-L897A-F900A mEm-CTNNA1 constructs and exposed to BafA1 treatment (400 nM, 6 h). GFP/GAPDH densitometry is shown on the right: mean ± s.d. (n = 5 independent experiments; *P < 0.05, ns not significant; two-tailed one sample t-test). **b** Representative immunoblot of MCF10A cells expressing Flag-wild-type and the indicated CTNNA1 mutants upon BafA1 treatment (400 nM, 6 h). **c** Representative immunoblots of MCF10A cells transiently expressing wild-type or Y419A-V422A mEm-CTNNNA1 and exposed to autophagy inducers: SMER28 (20 μM) and EBSS for 6 h. The graphs show the GFP/GAPDH densitometry of one representative experiment: mean ± s.d. (n = 3 independent experiments; **P < 0.01, *P < 0.05, ns not significant; two-tailed one sample t-test). The experiment was repeated with similar results. **d** Representative immunoblot of MCF10A cells expressing Flag-wild-type and the indicated CTNNA1 mutants upon exposure to different concentrations of SMER28 (20 and 50 μM) for 24 h. **e** Representative immunoblot of MCF10A cells expressing Flag-wild-type and the indicated CTNNA1 mutants upon exposure to EBSS for 4 h. Bars represent the mean ± s.e.m. (n = 4 independent experiments; *P < 0.05, ns not significant; two-tailed one sample t-test). Exact P values for asterisks: **a** (from left to right) 0.0317, 0.1743, 0.4750, 0.3386; **c** (from left to right) 0.0496, 0.0096, 0.5166, 0.1364; **e** (from left to right) 0.0336, 0.9923, 0.3824, 0.4799.

---

Healthcare), anti-rabbit (NA934V, GE Healthcare) and anti-goat (no. 611620, Invitrogen-Life Technologies); the following LI-COR secondary antibodies were used: anti-mouse 680 and anti-rabbit 800. F-actin staining was detected with Phalloidin-Alexa Fluor 546 (A22283, Invitrogen-Life Technologies) and 488 (A12379, Invitrogen-Life Technologies).

**Plasmids and siRNAs**. The following constructs were used in this study: 8XGTIIC-luciferase (no. 34615, Addgene)[32], pCMV-Renilla Luciferase (E2261, Promega), empty mEmerald-C1 (no. 54734, Addgene), mEmerald-Alpha1-Catenin-C-18 (no. 53982, Addgene), GFP-YAP (no. 17843, Addgene). The mRFP-LC3 construct was kindly provided by Dr. Tamotsu Yoshimori (Osaka University, Japan). Alpha1-Catenin which was amplified from mEmerald-Alpha1-Catenin-C-18 was cloned into pFlag-CMV-5a expression vector (E7523, Sigma-Aldrich) using EcoRI and SalI restriction enzyme sites (pFlag-CTNNA1).

Pre-designed siRNAs (On-Target plus SMART pool and/or set of deconvoluted oligos) targeting Control (D-001810-10), ATG7 (L-020112-00), ATG10 (L-019426-01), ATG16L1 (L-021033-01), CTNNA1 (L-010505-00) and CTNNA3 (L-020300-00) were purchased from Dharmacon Thermo Scientific.

(1) set of deconvoluted oligos for ATG7/10 #1 (J-020112-05, J-019426-09):
ATG7 siRNA_05: CCAACACACUCGAGUCUUU
ATG10 siRNA_09: CGUCUCAGGAUGAACGAAA
(2) set of deconvoluted oligos for ATG7/10 #2 (J-020112-06, J-019426-10):
ATG7 siRNA_06: GAUCUAAAUCUCAAACUGA
ATG10 siRNA_10: AGGAAUUGCGGCACGAAGA
(3) set of deconvoluted oligos for ATG7/10 #3 (J-020112-06, J-019426-11):
ATG7 siRNA_06: GAUCUAAAUCUCAAACUGA
ATG10 siRNA_11: GGAGGAGGCUUUCGAGCUA
(4) set of deconvoluted oligos for ATG7/10 #4 (J-020112-06, J-019426-12):
ATG7 siRNA_06: GAUCUAAAUCUCAAACUGA
ATG10 siRNA_12: CCAACGUUAUUGUGCAGAA

**Mutagenesis**. CTNNA1 mutants (Y419A-V422A, F511D-V514D, Y619A-I622A, L879A-F900A or Y419A-V422A- L879A-F900A) were generated using the Quickchange site-directed mutagenesis kit (Agilent Technologies, cat# 200514) according to the manufacturer's instructions. Mutagenesis primers were designed using web-based QuikChange Primer Design programme (Agilent Technologies). The primer sequences were used for mutagenesis are displayed in Supplementary Table 1. PCR products were incubated with Dpn1 restriction enzyme for 1 h and then mixed with XL-10 Gold-competent cells. After transformation, DNA was prepared from colonies and sequenced by Genewiz (UK). After establishing a CTNNA1 Y419A-V422A mutant, CTNNA1 mutant (CTNNA1 Y419A-V422A) was used to generate a CTNNA1 Y419A-V422A- L879A-F900A using mutagenesis primers for L897A-F900A.

**Reagents**. Bafilomycin A1 (Millipore) was dissolved with dimethyl sulfoxide (DMSO, Sigma-Aldrich) and used at either 400 nM for 4–6 h or 200 nM overnight. SMER28 was dissolved with DMSO and used at either 20 μM or 50 μM for 24 h. VPS34-IN1 (Vps34 Inhibitor, no. 532628, Calbiochem) was resuspended in DMSO and used at a concentration of 1 μM for 24 h. All control conditions without BafA1, SMER28 or VPS34-IN1 received an equivalent volume of DMSO as the compounds. Tat-Beclin1 L11S Peptide (Tat-scrambled control, NBP-49887) and Tat-Beclin1 D11 Autophagy Inducing Peptide (Retroinverso form, NBP-49888) were obtained from Novus Biologicals. Both peptides were dissolved with distilled water (DW) and used at 20 μM concentration for 48 h. Trehalose (D-(+)-Trehalose dihydrate, no. T9351, Sigma-Aldrich) was dissolved with DW and used at 100 mM for 24 h.

**Cell culture**. MCF10A cells were obtained from Horizon (no. HD PAR-058) and were cultured in DMEM:F12 (D6421, Sigma-Aldrich) media containing 5% horse serum (H1270, Sigma-Aldrich), 2 mM L-glutamine (G7513, Sigma-Aldrich) and 100 U/ml penicillin–streptomycin (P0781, Sigma-Aldrich) and supplemented with

0.1 μg/ml cholera toxin (C8052, Sigma-Aldrich), 20 ng/ml hEGF (E9644, Sigma-Aldrich), 10 μg/ml insulin (I9278, Sigma-Aldrich), 0.5 μg/ml hydrocortisone (H0135, Sigma-Aldrich), as previously reported[21].

Human cervical epithelium (HeLa) cells, human embryonic kidney 293T (HEK293T) cells, human breast cancer MCF7 cells, and human hepatocytes THLE2 cells were obtained from American Type of Cell Collection (ATCC, USA) and human lung carcinoma A549 cells were kindly provided by Dr. F. Buss (University of Cambridge, UK). HeLa cells, HEK293T, MCF7 and A549 cells were cultured in DMEM (D6546, Sigma-Aldrich) containing 10% foetal bovine serum (FBS, F7524, Sigma-Aldrich), 100 U/ml penicillin–streptomycin (P0781, Sigma-Aldrich), 2 mM L-glutamine (G7513, Sigma-Aldrich). THLE2 cells were cultured in Bronchial Epithelial Cell Growth Medium (BEGM) supplemented with Bullet kit (BEGM Bullet Kit; CC3170, Lonza/Clonetics Corporation) added extra 5 ng/mL EGF, 70 ng/mL Phosphoethanolamine and 10% foetal bovine serum. Human liver carcinoma HepG2 cells were purchased from European Collection of Authenticated Cell Cultures and cells were grown in RPMI-1640 (R0883 Sigma-Aldrich) containing 10% FBS (F7524, Sigma-Aldrich), 100 U/ml penicillin–streptomycin (P0781, Sigma-Aldrich), 2 mM L-glutamine (G7513, Sigma-Aldrich).

All the cell lines were incubated at 37 °C and 5% $CO_2$, humidified atmosphere and were tested for mycoplasma contamination every 2 weeks.

**Primary mouse mammary epithelial cells (pMECs)**. The study complied with all relevant ethical regulations for animal testing and research. All studies and procedures were performed under the jurisdiction of appropriate UK Home Office Project and Personal animal licences and with the approval of the University of Cambridge Animal Welfare and Ethical Review Body. The mice were housed in individually ventilated cages with free access to standard animal food chow and water, in a climate-controlled room with a 12 h light/dark cycle.

Isolation of pMECs followed the protocol described in Pavel et al.[21]. Briefly, at E16.5 gestation, female mice (wild-type BL6/BC122 or Atg16L1 hypomorph mice[12]) were sacrificed by cervical dislocation and the abdominal and inguinal mammary glands were collected in HBSS. Mammary glands were next transferred in a Petri dish, minced with curved scissors (around 200 cuts) and incubated in 10 ml of DMEM:F12, containing 5% horse serum, 3 mg of Collagenase (C9891, Sigma-Aldrich) and 2.5 mg (around 1000 U) hyaluronidase (H3506, Sigma-Aldrich), on a rotating surface at 37 °C and 5% $CO_2$, humidified atmosphere for 2 h. The solution was next transferred to a 15 ml falcon tube and centrifuged at 500 × g for 5 min to pellet the cells, while the supernatant with the fat clumps were removed. The cells were resuspended twice in 10 ml red blood lysis buffer (at room temperature—RT for 5 min) and then incubated with DMEM (D6546, Sigma-Aldrich) supplemented with 10% FBS (F7524, Sigma-Aldrich), 100 U/ml penicillin–streptomycin (P0781, Sigma-Aldrich), 2 mM L-glutamine (G7513, Sigma-Aldrich) in a T-75 flask at 37 °C and 5% $CO_2$, humidified atmosphere for 1 h, to allow the majority of the fibroblasts to attach to the bottom of the T-75 culture flask. The suspension of epithelial cells was transferred to a 15 ml falcon tube and incubated in 0.25% pre-warmed trypsin-EDTA (25200, Invitrogen) at RT for 5 min. The trypsin reaction was stopped with 10 ml of DMEM:F12, containing 5% horse serum and 10 g/l DNase (DN25, Sigma-Aldrich) and the epithelial cells were pelleted at 500 × g, resuspended and cultured in the MCF10A culture medium: DMEM:F12 (D6421, Sigma-Aldrich) supplemented with 5% horse serum (H1270, Sigma-Aldrich), 0.1 μg/ml cholera toxin (C8052, Sigma-Aldrich), 20 ng/ml hEGF (E9644, Sigma-Aldrich), 10 μg/ml insulin (I9278, Sigma-Aldrich), 0.5 μg/ml hydrocortisone (H0135, Sigma-Aldrich), 2 mM L-glutamine (G7513, Sigma-Aldrich) and 100 U/ml penicillin–streptomycin (P0781, Sigma-Aldrich) at 37 °C and 5% $CO_2$, humidified atmosphere.

**Primary mouse cortical neurons (p. neurons) and primary mouse embryonic fibroblasts (pMEFs)**. Mouse primary cortical neurons were isolated and cultured as described by Jimenez-Sanchez et al.[35]. Dr T. Ricketts helped with the isolation of cortical neurons from E16.5 mice. The neurons were plated in a 12-well plate. The plates were pre-coated with 2 ml of 20 μg/ml poly-L-ornithine hydrobromide (P3655, Sigma-Aldrich) in phosphate-buffered saline (PBS), overnight at 37 °C.

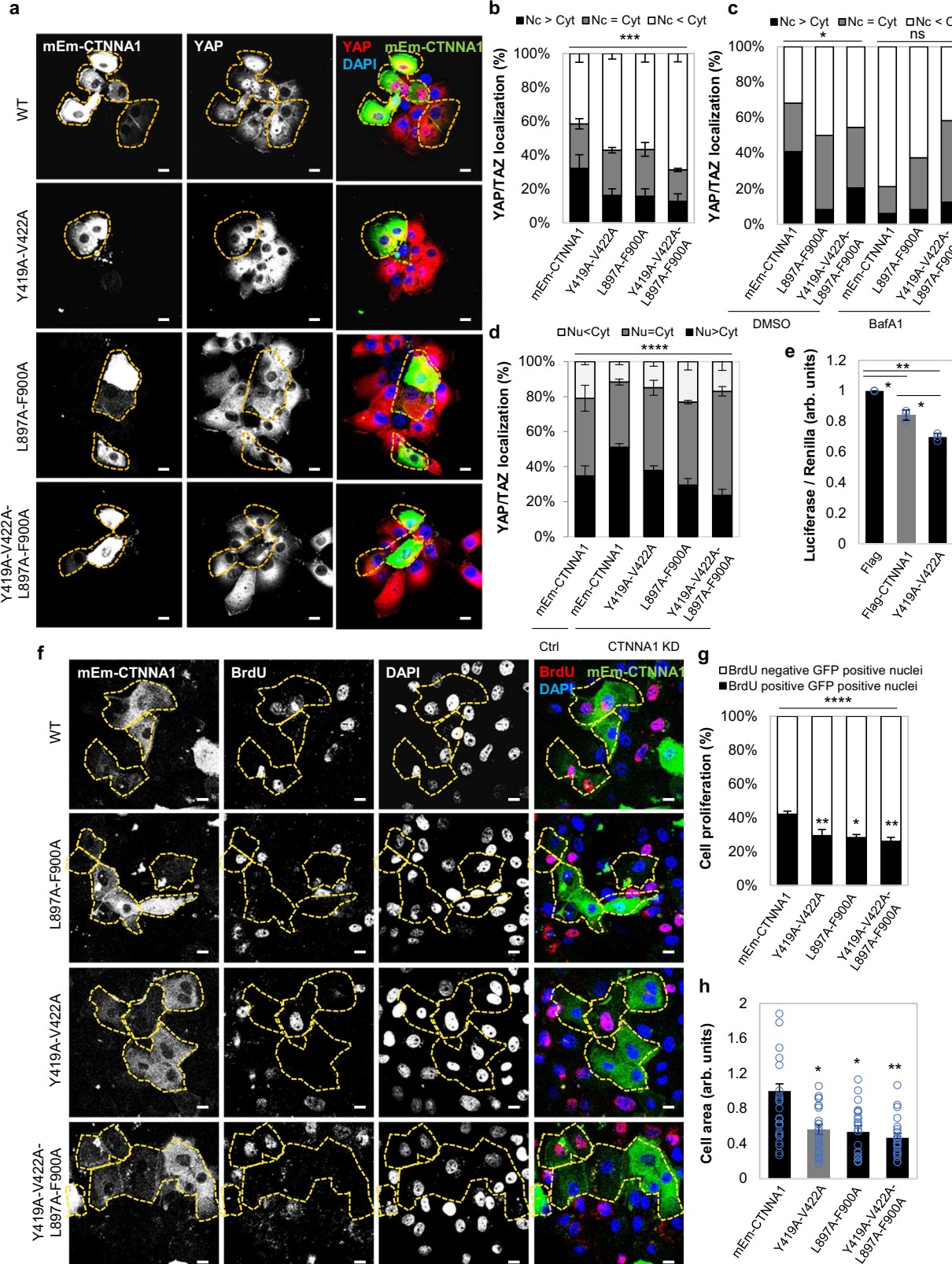

The heads of the E16.5 embryos were cut and washed in PBS-glucose 0.6%. Each head was held with forceps pinning the eyes, the back of the head (1/3 external, cerebellum) was cut with small scissors and then the brain was pulled out gently with curved scissors. Then, the midbrain and olfactory bulb were removed, the brain cut into two hemispheres, the meninges peeled off and the striatum cut off. The cortex was transferred in a new Petri dish containing PBS-glucose 0.6% and minced with forceps. The minced cortex was next transferred to a 15 ml falcon tube and homogenised 20 times with a 1 ml pipette. The viable neurons were pelleted by centrifugation, resuspended in culture medium and seeded on pre-coated plates. The medium was changed after 3–4 h. Neurons were cultured in Neurobasal-A medium (12349-015, GIBCO Life technologies) supplemented with 200 mM B27 (17504-044, GIBCO Life Technologies), 2 mM L-glutamine (G7513, Sigma-Aldrich) and 100 U/ml penicillin–streptomycin (P0781, Sigma-Aldrich) at 37 °C and 5% $CO_2$, humidified atmosphere. One half of the culture medium was

**Fig. 8 Expression of LIR-defective CTNNA1 mutants inhibits YAP/TAZ activity. a** Representative confocal images of YAP/TAZ endogenous immunostaining in MCF10A cells expressing either wild-type or mutant (Y419A-V422A, L897A-F900A or Y419A-V422A-L897A-F900A) mEm-CTNNA1. Scale bars are 10 μm. The experiment was repeated three times with similar results. **b** YAP/TAZ localisation in MCF10A cells transfected with wild-type or mEm-CTNNA1 mutants. Bars represent the mean ± s.e.m. (*n* = 4 independent experiments; ***P* < 0.001; two-way ANOVA). **c** YAP/TAZ localisation in MCF10A cells transfected with wild-type or mutants of mEm-CTNNA1 upon BafA1 exposure (200 nM, 16 h). Around 100 cells were analysed per condition —*P* < 0.05, ns not significant; chi-squared test. **d** YAP/TAZ localisation in MCF10A cells transfected with wild-type or mutant forms of mEm-CTNNA1 upon α-catenin knockdown. Bars represent the mean ± s.d. (*n* = 3 independent experiments; ****P* < 0.0001; two-way ANOVA). **e** TEAD luciferase activity in MCF10A cells expressing either wild-type or mutant (Y419A-V422A) Flag-CTNNA1. Bars represent the mean ± s.d. (*n* = 3 independent experiments; ***P* < 0.01, **P* < 0.05; two-tailed one sample *t*-test). **f** Representative confocal images of BrdU-positive MCF10A cells transfected with wild-type or mutant forms of mEm-CTNNA1. Scale bars are 10 μm. The experiment was repeated three times with similar results. **g** Percentages of BrdU-positive cells in MCF10A cells transfected with wild-type or mutant forms of mEm-CTNNA1. Bars represent the mean ± s.e.m. (*n* = 4 independent experiments; ****P* < 0.001, ***P* < 0.01, **P* < 0.05; two-way ANOVA). **h** Size of MCF10A cells expressing either wild-type or the indicated mEm-CTNNA1 mutants. *n* = 25 cells per condition were analysed for measuring cell area using ZEN software. Bars represent the mean ± s.e.m (***P* < 0.01, **P* < 0.05; two-tailed *t*-test). Exact *P* values for asterisks: **b** 0.0003; **e** (from left to right) 0.0146, 0.0022, 0.0049; **g** (from left to right) 0.0043, 0.0207, 0.0019; **h** (from left to right) 0.0153, 0.0102, 0.0010.

---

replenished with pre-warmed fresh medium every 2–3 days. At DIV 7, the differentiated cortical neurons were harvested, lysed and analysed by western blotting.

pMEF were previously described[21,36]. Briefly, E16.5 mice were sacrificed and embryos were harvested. The primary mouse fibroblasts were grown in DMEM (D6546 Sigma-Aldrich) supplemented with 10% foetal bovine serum (FBS, F7524 Sigma-Aldrich), 100 U ml$^{-1}$ penicillin–streptomycin (P0781 Sigma-Aldrich), 2 mM L-glutamine (G7513 Sigma-Aldrich) at 37 °C and 5% CO$_2$, humidified atmosphere.

**3D culture of MCF10A cells**. MCF10A cells and pMECs were cultured in Collagen-Matrigel 3D matrix, as described previously[21]. Briefly, the 3D extracellular matrix (ECM) was made of mixing Growth Factor Reduced Matrigel (BD Biosciences) (5% of the final volume) and Rat Collagen I (TREVIGEN), 1 mg/ml (soft matrix) and 3 mg/ml (stiff matrix) final concentrations. Collagen I was initially neutralised with 7.5% sodium bicarbonate in 1× PBS according to the manufacturer's instructions. 1× volume of ECM was combined with 1× cold medium and used for coating Nunc 8 chamber slides (150 μl of the mixture was added per well). After the ECM gelled at 37 °C and 5% CO$_2$, a mixture of 1× volume ECM with 1× volume of cells (2000 cells in 100 μl) was added (in drops) on the top of the pre-gelled ECM (150 μl of final mixture per well). The wells were replenished every 2 days with the appropriate culture medium.

**Stable isotope labelling by amino acids in cell culture (SILAC)-based quantitative proteomics**. HeLa cells were grown in SILAC DMEM medium (88364, ThermoFisher Scientific) supplemented with 10% dialysed foetal calf serum (FCS) (26400, ThermoFisher Scientific), 100 U/ml penicillin–streptomycin (P0781, Sigma-Aldrich). Media was supplemented with either light (Arg 0, Lys 0, Sigma-Aldrich), medium (Arg 6, Lys 4, Cambridge Isotope Laboratories, Andover, MA) or heavy (Arg 10, Lys 8, Cambridge Isotope Laboratories, Andover, MA) amino acids at 50 mg/l and l-proline at 280 mg/l. The cells were kept in culture for 6 days before treatment with BafA1 (200 nM, 24 h). The cells were then washed twice in PBS, collected in LoBind microcentrifuge tubes (0030122356, Eppendorf) and lysed in 200 μl of lysis buffer (2% SDS, 100 mM Tris/HCl pH 7.5, 1× Roche protease inhibitor cocktail) for 5 min at RT. The samples were sonicated 10 × 10 s bursts of medium-high power (on a Diagenode Bioruptor) to shear DNA, centrifuged at 16,000 × *g* for 5 min and equal amount of either M/L (DMSO/ BafA1) or H/M (DMSO/ BafA1) cell lysates were combined to proceed to protein extraction and digestion, off-line peptide fractionation by high-pH reverse-phase high-pressure liquid chromatography (HpRP-HPLC) and mass-spectrometry analysis, following the protocol described in ref. [37,38]. Briefly, data for SILAC samples were generated using an Q Exactive Orbitrap mass spectrometer coupled to an RSLCnano3000 (Thermo Scientific). Raw MS files were processed using MaxQuant 1.3 and data were searched against a human Uniprot database (downloaded 03/06/14, 20,176 entries)—see Supplementary Data 1.

**Transfection**. For siRNA transfection, MCF10A, HEK293T, HeLa and HepG2 cells were plated in six-well plates and cells were transfected with 100 nM siRNA using Lipofectamine RNAiMAX (13778150 Invitrogen) for 6 h per well. The following day, cells were re-transfected with 50 nM siRNA. When required, cells were split in the 6- or 12-well plates.

For DNA transfection, MCF10A cells were seeded in six-well plates and cells were transfected with 1 μg of DNA constructs (CTNNA1 wild type or mutants) using TransIT-2020 reagent (Mirus). HepG2 cells were cultured in six-well plate and transfected with wild-type CTNNA1 (1 μg) by using TransIT-2020 reagent (Mirus). After 24 h, cells were re-transfected with 1 μg of cDNA in order to increase the transfection efficiency. HeLa and MCF7 cells were seeded in six-well plates and cells were co-transfected with 1 μg of both mRFP-LC3 and mEm-CTNNA1 using

TransIT-2020 reagent (Mirus). On the following day, cells were reseeded in the 6- or 12-well plates according to the experimental requirements.

**IF microscopy**. For IF staining, cells were cultured on coverslips. After treatment of experimental requirements, cells were washed with PBS three times and then fixed with 4% paraformaldehyde (PFA, Sigma-Aldrich) for 5 min. After rinsing the cells on coverslips, cells were permeabilised with 0.1% Triton X-100 for 10 min and blocked with 3% BSA (BP1605-100, Fischer Scientific) for 1 h. Cells were incubated with primary antibodies in blocking buffer overnight at 4 °C. After washing three times with PBS, cells were incubated with secondary antibodies tagged with Alexa Fluor obtained from molecular Probes for 90 min at RT. Cells on coverslips were washed with PBS and mounted in Prolong Gold Antifade reagent with DAPI (P-36931, Invitrogen).

For immunostaining of LC3B, cells were fixed and permeabilized with ice-cold methanol for 4 min. After washing three times with PBS, cells were blocked with 3% BSA for 1 h. The following steps were the same as above.

The images were performed by LSM710, LSM780 and LSM880 confocal microscopy (40×, 63× NA 1.4 Plan Apochromat oil-immersion lens, Carl Zeiss). Further analysis was performed in Photoshop (Adobe) and ImageJ, and if required, equal adjustments were made for all the images from all control and treatment groups.

**BrdU (5-bromo-2′-deoxyuridine) staining**. For cell proliferation assay, cells were seeded and grown on coverslips. Freshly prepared BrdU (5-bromo-2′-deoxyuridine, ab142567) was diluted to 10 μM in pre-warmed culture media and added to the cells for 2 h. Then, the media was removed and cells were fixed in 4% PFA for 30 min and incubated with 1.5 N HCl for 1 h. Then, cells were washed three times in PBS and blocked with 5% goat serum and 0.3% TritionX-100 in 1× PBS for 90 min. After, cells were incubated with primary anti-BrdU antibody (1:200 dilution in PBS containing 5% goat serum), overnight at 4 °C. Cells were rinsed with PBS for three times and incubated with secondary antibody (at 1:400 dilution) for 1 h at RT. After washing with PBS, the coverslips were mounted with the Prolong Gold Antifade reagent with DAPI (P-36931 Invitrogen). At least 500 cells were counted for each condition in order to determine the percentage of BrdU-positive cells. For counting BrdU-incorporated mEm-CTNNA1 cells, cells were transfected with either wild-type or mutants of mEm-CTNNA1. On the following day, transfected cells were incubated with BrdU solution for 24 h. The BrdU solution was discarded and then cells were fixed with 4% PFA for 15 min at RT. Then, cells were incubated with 1 N HCl on ice for 10 min, followed by 2 N HCl for 10 min at RT. The following procedures were the same as above[21]. At least 150 cells were counted for each condition in order to determine the percentage of BrdU-mEmerald positive cells.

**Live-cell imaging**. MCF7 and HeLa cells were seeded at a density of ~1.5 × 10$^5$ cells per dish and transfected with 1 μg mEm-CTNNA1 and 0.5 μg mRFP-LC3 using Mirus Bio *Trans*IT ®-2020, per well of a six-well plate. Cells were placed in EBSS with HEPES (4-(2-hydroxyethyl)-1-piperazineethanesulfonic acid) and immediately after, imaged at 37 °C on an incubated Zeiss AxioObserver Z1 microscope with a LSM780 confocal attachment using a 63× 1.4 NA Plan Apochromat oil-immersion lens.

**Cell lysis and western blot analysis**. Cells were seeded and cultured in 6- or 12-well plates. Cells were washed twice in cold PBS and lysed in either RIPA buffer (150 nM NaCl, 1% NP-40, 0.5% NaDoc, 0.1% SDS, 50 mM Tris/HCl pH 7.4—all from Sigma-Aldrich) containing protease and phosphatase inhibitors (from Roche Diagnostics) or directly in 1× Laemmli Sample buffer (diluted from 2× Laemmli Sample Buffer #161-0737 BioRad), on ice. The samples lysed in RIPA buffer were

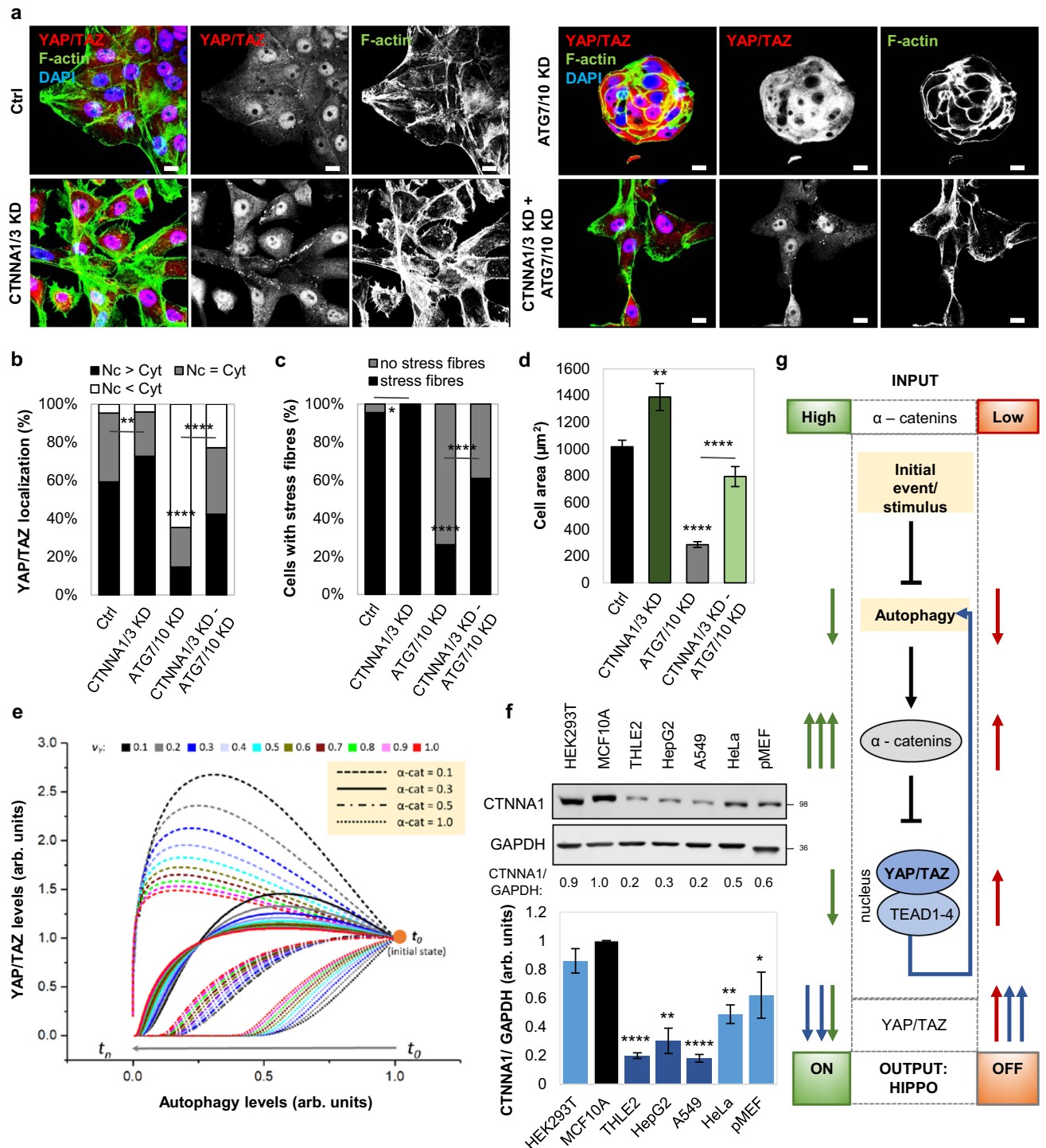

incubated 30 min on ice to ensure complete lysis, and then centrifuged at $16,000 \times g$ for 10 min at 4 °C, to pellet the cellular debris. The supernatant of each sample was collected in a separate E-tube and diluted with 2× Laemmli Sample Buffer (161-0737 BioRad) and boiled for 10 min at 100 °C. For WB analysis, samples were subjected to an SDS-PAGE separation and transferred on PVDF membranes. PVDF membranes were blocked with 4% skim-milk in PBS containing 0.1% Tween for 1 h and incubated with primary antibody at 4 °C overnight. After washing with 0.1% Tween-PBS, the membranes were incubated with secondary antibody for 90 min at RT and subjected to protein visualising using either LI-COR Odyssey scanner or the ECL enhanced chemiluminescence detection kit from GE Health-care. Protein levels detected by ECL were quantified using ImageJ software, while the protein levels detected by IRdye were quantified using Image Studio Software (LI-COR).

**Cytoplasmic/nuclear fractionation.** Nuclear and cytoplasmic fractionation was performed using NE-PER Nuclear and Cytoplasmic Extraction Kit (Cat# 78833 Thermofisher), according to the manufacturer's instructions. Briefly, cells were washed twice with ice-cold PBS and centrifuged at $500 \times g$ for 5 min. The cell pellet was resuspended in 200 µl of lysis buffer (CER I), followed by 10 min incubation on ice. After adding 11 µl of CERII, samples were vortexed for 10 s and incubated for 1 min on ice. For collecting cytoplasmic extracts, samples were centrifuged at $16,000 \times g$ for 15 min and the supernatants (cytoplasmic extracts) were collected to pre-chilled new tubes. The insoluble cell pellets, containing crude nuclei, were resuspended in Nuclear Extraction Reagent, incubated on ice and vortexed for 15 s every 10 min (up to 4 times). For extracting nuclear fraction, cells were centrifuged at $16,000 \times g$ for 10 min and supernatants (nuclear extracts) were transferred to pre-chilled new tubes. The fractionated samples were mixed with 2× Laemmli

**Fig. 9 Autophagy regulates YAP/TAZ activity via α-catenins. a** Representative confocal images of endogenous YAP/TAZ and F-actin immunostaining in MCF10A cells. MCF10A cells were initially exposed to ATG7/10 siRNAs for 48 h, and only after, followed by α-catenin depletion using CTNNA1/CTNNA3 siRNAs (together with ATG7/10 siRNAs) for other 48 h. Scale bars are 10 μm. The experiment was repeated with similar results. **b** YAP/TAZ localisation in MCF10A cells treated as in (**a**). CTNNA1/3 KD partially rescues the YAP/TAZ localisation phenotype in ATG7/10 KD cells. The experiment was repeated with similar results (n (from left to right) = 280/304/299/333 cells; \*\*P < 0.01, \*\*\*\*P < 0.0001; chi-square test). **c** Quantification of percentages of MCF10A cells with F-actin stress fibres. MCF10A cells were initially exposed to ATG7/10 siRNAs for 48 h, followed by α-catenin depletion using CTNNA1/CTNNA3 siRNAs for other 48 h. The experiment was repeated with similar results (n (from left to right) = 202/ 200/ 206/ 233 cells; \*P < 0.05, \*\*\*\*P < 0.0001; chi-square test). **d** Size of MCF10A cells initially exposed to ATG7/10 siRNAs for 48 h, followed by α-catenin depletion using CTNNA1/CTNNA3 siRNAs for other 48 h. Confocal images of 25 cells per condition were analysed for measuring cell area using ZEN software. Bars represent the mean ± s.e.m (\*\*\*\*P < 0.0001, \*\*P < 0.01; two-tailed t-test). **e** Mathematical modelling of YAP/TAZ activity when reducing autophagy levels. The graph shows the dependency of YAP/TAZ activity on the initial levels of α-catenins and the strength of the feedback loop involving YAP/TAZ control of autophagy $v_y$[21]. **f** Representative immunoblot of α-catenin in various cell lines. Bars represent the mean ± s.d. (n = 3 independent experiments; \*\*\*P < 0.001, \*\*P < 0.01, \*P < 0.05; two-tailed one sample t-test). **g** Model for the autophagy-dependent YAP/TAZ activity in two initial conditions: low and high α-catenin levels. Autophagy inhibition in cells with high initial α-catenin levels promotes α-catenin accumulation which sequester YAP/TAZ into the cytosol (Hippo signalling is ON). However, autophagy inhibition in cells with low initial α-catenin levels causes only a small increase in these proteins, increase unable to sequester YAP/TAZ into the cytosol (as their levels also increase, being also autophagy substrates)—Hippo signalling is OFF. The feedback loop of YAP/TAZ controlling autophagy accentuates these effects. Exact P values for asterisks: **b** 0.0023; **c** 0.0105; **d** 0.0082; **f** (from left to right) <0.0001, 0.0027, <0.0001, 0.0027, 0.0276.

---

buffer. LaminB was used as loading control for nuclear fractions, while S6 and GAPDH were used as cytosolic loading controls.

**Cytoplasmic fractionation.** Cytoplasmic fractionation was performed according to the method described by Baghirova et al. [39]. Briefly, HEK293T cells were co-transfected with 4 μg of Flag-tagged constructs (empty, CTNNA1 wild type or LIR mutants) together with 4 μg of YAP-GFP in 10 cm dishes. After 24 h, cells were treated with DMSO or BafA1 200 nM overnight or incubated with or without EBSS for 6 h. Cells were collected and centrifuged at 500 × g for 5 min at 4 °C. Cell pellets were washed with 1 ml of ice-cold PBS and 50 μl of cells were further mixed with 2× Laemmli Sample buffer as total input samples. The rest of the cells were centrifuged at 500 × g for 10 min at 4 °C and the supernatant was discarded. The cell pellet was resuspended in lysis buffer (NaCl 150 mM; HEPES (pH 7.4) 50 mM; Digitonin (D141, Sigma-Aldrich) Hexylene glycol (12100, Sigma-Aldrich)) with protease inhibitors and incubated on rotator for 10 min at 4 °C. Lysed cells were centrifuged 2000 × g for 10 min at 4 °C and the supernatant was collected. Before the immunoprecipitation, 50 μl of cytoplasmic extracts were mixed with 2× Laemmli Sample buffer and used as input cytoplasmic samples. All samples were boiled for 10 min at 4 °C. Cytoplasmic fractionation was performed using the Flag-trap method.

**Luciferase reporter assay.** Luciferase reporter assay was performed using the Dual-Glo luciferase assay system kit (E1910 Promega), following the protocol previously described[21]. Briefly, cells were co-transfected with 1 μg of luciferase reporter (8XGTIIC) together with 0.1 μg of renilla luciferase for 24 h per well in six-well plates. Transfected cells were lysed in 200 μl of 1× Passive lysis buffer for 15 min at RT and collected by scraping in E-tubes. After centrifugation at 16,000 × g for 10 min, 20 μl of supernatant from each lysed sample were transferred in triplicate in a 96 wells microplate and mixed with 100 μl of Luciferase Assay Reagent II (LARII). The firefly luciferase activity was measured using a SPARK multimode microplate reading Luminometer (TECAN Trading AG). The renilla luciferase activity was measured immediately after adding 100 μl of Stop & Glo reagent to the reaction.

MCF10A and HepG2 cells were transfected with either 1 μg CTNNA1-Flag construct (either wild-type or the indicated CTNNA1 mutants) for 24 h. In addition, the cells were co-transfected with 1 μg of luciferase reporter (8XGTIIC) together with 0.1 μg of Renilla luciferase for 24 h and cultured in full medium. The following procedures were the same as described above. The TEAD activity was expressed as the ratio between firefly and Renilla luciferase values.

**Immunoprecipitation.** For immunoprecipitation of mEmerald-tagged proteins using the GFP-Trap method (gtma-100, ChromoTek), cells were lysed with lysis buffer (10 mM Tris/Cl pH 7.5; 150 mM NaCl; 0.5 mM EDTA; 0.5% NP-40) containing protease and phosphatase inhibitors from Roche Diagnostics for 30 min at 4 °C. The lysed cells were centrifuged at 13,000 × g for 10 min and the supernatant transferred to a new E-tube and mixed with dilution buffer (10 mM Tris/HCl pH 7.5; 150 mM NaCl; 0.5 mM EDTA). For each sample, 50 μl of supernatant were mixed with 2× Laemmli buffer, boiled at 100 °C for 10 min and used for input loading. The samples were further incubated with pre-washed 25 μl of GFP-Trap beads for 1 h at 4 °C on a rotating surface. GFP-beads were washed three times with dilution buffer, resuspended with 2× Laemmli buffer and boiled at 100 °C for 10 min.

Immunoprecipitation of Flag-tagged proteins was performed with anti-Flag M2 Magnetic Beads (M8823, Sigma-Aldrich). Cells were lysed with NETN lysis buffer (20 mM Tris/Cl pH 7.5; 100 mM NaCl; 0.5 mM EDTA, 0.5% NP-40, supplemented with protease and phosphatase inhibitors cocktails (Roche)) for 30 min at 4 °C. The lysed cells were centrifuged at 16,000 × g for 10 min and then the supernatants were collected into new E-tubes. For each sample, 50 μl of supernatant were mixed with 2× Laemmli buffer, boiled at 100 °C for 10 min and used for input loading. The samples were mixed with anti-Flag M2 Magnetic Beads and incubated at 4 °C overnight on a rotating surface. After five times washing, Flag beads were denatured with 2× Laemmli buffer and boiled at 100 °C for 10 min.

For immunoprecipitation of endogenous protein, cells were incubated with ice-cold NETN buffer for 30 min at 4 °C. After centrifugation of lysed cells at 16,000 × g for 10 min, supernatants were collected into separate tubes and incubated with primary antibodies at 4 °C overnight. The following day, the immune-complexes were cleared with 20 μl of Dynabeads-Protein G (10004D, Life technologies) for 2 h at 4 °C and then, purified by boiling in 2× Laemmli buffer for 10 min.

Input lysates were run simultaneously with the immunoprecipitation samples on 15% polyacrylamide gels and visualised on an Odyssey scanner.

**Purification of recombinant LC3.** LC3 was purified as described by Puri et al.[40]. Briefly, LC3B was expressed using a pGEX-6p1 plasmid in *Escherichia coli* for 3 h at 37 °C after induction with 1 mM IPTG. Cells were lysed in 50 mM Tris pH 7.4, 150 mM NaCl using a sonicator in the presence of proteases inhibitors and centrifuged at 180,000 × g for 30 min at 4 °C. The supernatant was incubated with glutathione-Sepharose 4B beads followed by extensive washes in lysis buffer. Pre-Scission Protease (27-0843-01, GE Healthcare) was added at 100 U/ml in a two-bed volume of PreScission Buffer (50 mM Tris/HCl pH 7.5, 150 mM NaCl, 1 mM EDTA) freshly prepared with 1 mM DTT and cleavage was performed overnight at 4 °C. Cleaved protein was eluted and stored at −80 °C.

**Protein binding assay.** Binding of LC3B (purified from *E. coli*) to CTNNA1 (TP301776, Origene) was performed by incubation of recombinant proteins at 25 °C for 1 h, followed by immunoprecipitation with rabbit anti-CTNNA1 antibody (ab51032, Abcam, 1:100) and processed for WB analysis.

**Structural analysis of α-catenin.** Visualisation and generation of graphic illustrations of the molecular models of human CTNNA1 (pdb. 4IGG)[4] were performed using the PyMOL Molecular Graphics System, Version 2.0 Schrödinger, LLC (http://www.pymol.org). The pdb. 4IGG [4] shows the structure of dimeric CTNNA1 (aminoacid sequences 82-878 and 82-861, respectively). As the 891-906 region is not included in the pdb. 4IGG [4], this amino acid sequence was computed using Phyre2. The amino acids shown in blue and green were mutated for each potential LIR region, while the amino acids coloured in orange–purple corresponds to the LIR regions that were not mutated (those amino acids are facing the interior of the CTNNA1 molecule and therefore they are unlikely to interact with LC3).

**Bioinformatic analysis of putative LIR regions.** Many Atg8/LC3/GABARAP-interacting proteins contain a basic hydrophobic LIR motif with the core consensus sequence (WFY) xx (ILV). The residues marked in bold (positions 3 and 6) correspond to the most crucial residues for the interaction with Atg8-family proteins[41]. The iLIR database (https://ilir.warwick.ac.uk), a freely available web resource[42], was used to identify the potential 6 LIR regions in the CTNNA1/2/3 amino acids sequences. For CTNNA1, the putative LIR regions correspond to: 146 DVYKLL 151, 175 IQYKAL 180, 243 LIYKQL 248, 417 KEYAQV 422, 509 DDFLAV 514, 617 LVYDGI 622. The LIR region (895 QALSEF 900) was

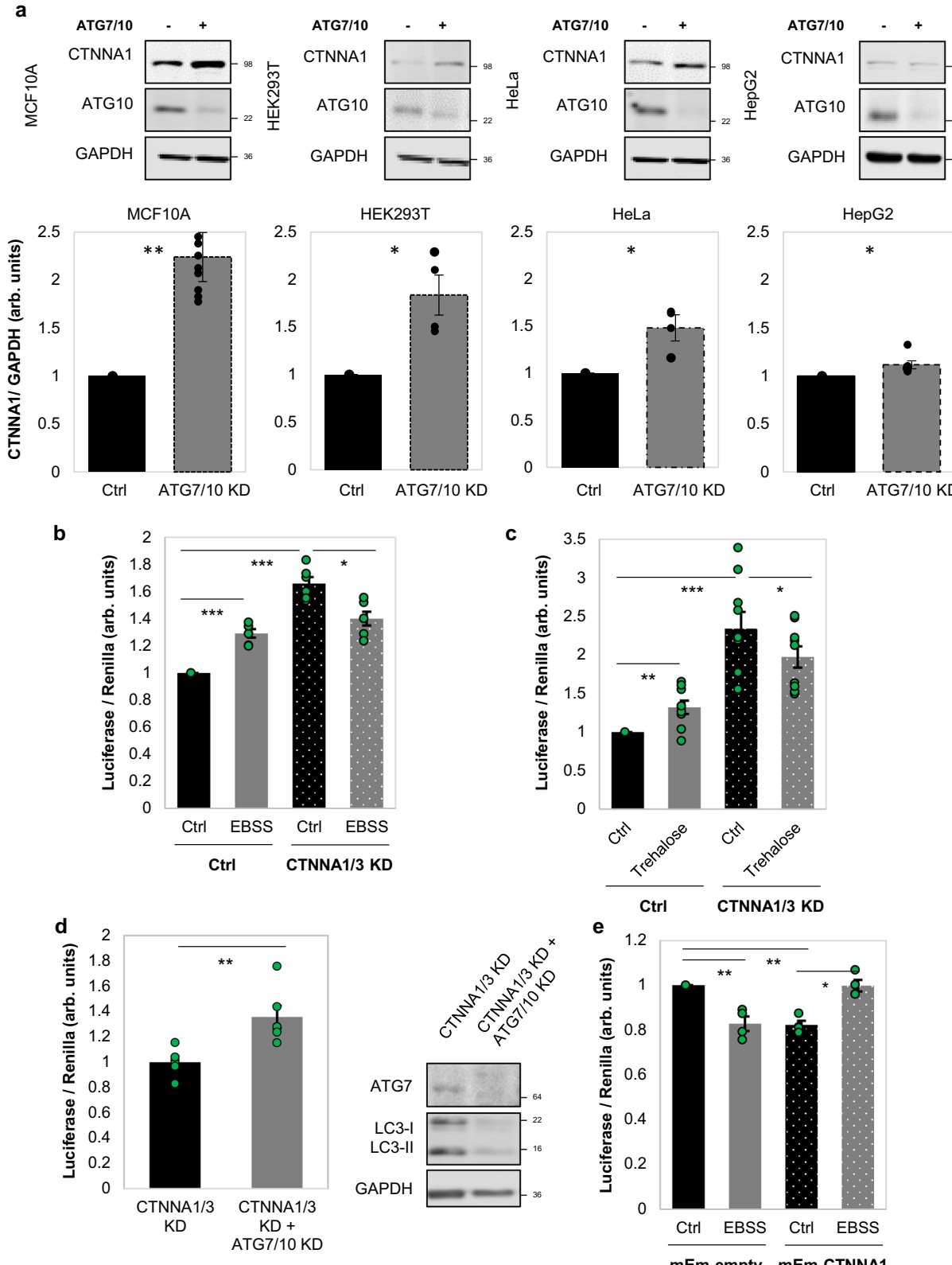

determined experimentally, as expression of the construct GFP-alpha-E-Catenin 3xFP4 (895) (no. 67936, Addgene)[43], which lacks the last 11 amino acids of the full length Ctnna1 (896 ALSEFKAMDSI 906), did not show accumulation within mRFP-LC3 vesicles by live-cell imaging (Supplementary Fig. 31).

**Statistical analysis**. Densitometry of western-blot bands was performed using either ImageJ (when using ECL-protein revealing) or Image Studio (for LI-COR

protein revealing). The graphs show the mean of independent experiments. One sample *t*-test (when the control was initially set up to 100 %), two-tailed Student's *t*-test, two-way ANOVA, chi-squared test and multinomial logistic regression were used to quantify *P* values in Origin v8.1 or SPSS Statistics v25.0. The statistical details of all experiments (including *n* numbers, error bars, *P* value statistical significance and the statistical test performed) are reported in the figure legends.

The colocalization Mander's and Pearson's coefficients were analysed using both ImageJ and Volocity softwares. Those experiments were examined three-

**Fig. 10 α-Catenin intracellular protein levels direct the YAP/TAZ response to autophagy perturbations. a** Representative immunoblots of CTNNA1 in multiple cell lines: MCF10A, HEK293T, HeLa, HepG2 after exposure to ATG7/10 siRNAs. Bars represent the mean ± s.e.m. ($n$ (from left to right) = 10/4/4/6 independent experiments; ***$P < 0.001$, *$P < 0.05$; two-tailed one sample $t$-test). **b** TEAD luciferase activity in MCF10A cells exposed to CTNNA1/3 siRNAs, followed by treatment with EBSS (6 h), as indicated. Bars represent the mean ± s.e.m. ($n = 6$ independent experiments; ***$P < 0.001$, *$P < 0.05$; two-tailed one sample $t$-test). **c** TEAD luciferase activity in MCF10A cells exposed to CTNNA1/3 siRNAs, followed by treatment with Trehalose (100 mM, 24 h), as indicated. Bars represent the mean ± s.e.m. ($n = 9$ independent experiments; ***$P < 0.001$, **$P < 0.01$, *$P < 0.05$; two-tailed one sample $t$-test). **d** TEAD luciferase activity in MCF10A cells initially exposed to CTNNA1/3 siRNAs for 48 h followed by double ATG7/10 KD for other 48 h, as indicated. Bars represent the mean ± s.e.m. ($n = 6$ independent experiments; **$P < 0.01$; two-tailed one sample $t$-test). **e** TEAD luciferase activity in HepG2 cells transfected with mEm-empty or wild-type mEm-CTNNA1, and after, exposed to EBSS for 6 h. Bars represent the mean ± s.e.m. ($n = 4$ independent experiments; **$P < 0.01$, *$P < 0.05$, ns not significant; two-tailed one sample $t$-test). Exact $P$ values for asterisks: **a** (from left to right) 0.0017, 0.0281, 0.0459, 0.0405; **b** 0.0001, <0.0001, 0.0160; **c** 0.0056, 0.0012, 0.0332; **d** 0.0046; **e** (from left to right) 0.0065, 0.0012, 0.0112.

independent repeats. Measurement of cell area and diameter of spherical cellular structures was determined using the ZEN 2.3 software.

**Mathematical-numerical model.** Our mathematical model is based on a set of three non-linear ordinary differential equations (ODEs), represented by autophagy levels ($A$), YAP/TAZ activity ($Y$) and α-catenin protein levels ($C$). Each term in the system of ODEs represents one single action.

The following basic assumptions were considered (see the scheme bellow):

(i) Reduction in YAP/TAZ activity leads to autophagy decrease via previously described mechanisms[21,31]. When initial YAP levels are high, the extent of autophagy inhibition is also high, and when initial YAP levels are low, there is still a loss of autophagy, but the effects are small.

(ii) YAP/TAZ activity results from the balance between two opposite mechanisms: (I) *reduction in YAP/TAZ activity* as a consequence of YAP/TAZ being sequestered into the cytosol by α-catenin (the LIR-dependent mechanism experimentally described herein), and (II) *increase in YAP/TAZ activity* as a consequence of perturbed autophagy (the mechanism experimentally described by Lee et al.[18]) —when initial autophagy is high, the increase in YAP resulting from autophagy inhibition is large, and when initial autophagy is low, there is still an increase in YAP, but marginal.

(iii) α-catenin protein levels accumulate upon autophagy perturbation (the mechanism experimentally described herein)—the rate of α-catenin accumulation is high when initial autophagy is high, and is low when autophagy is low.

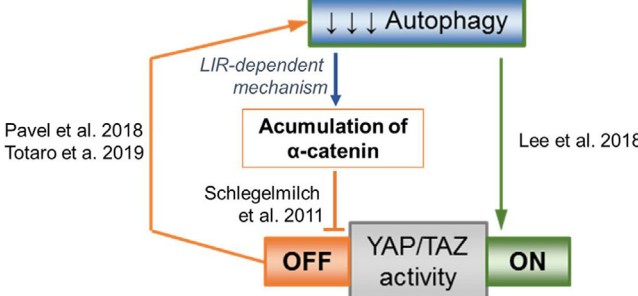

The ODEs of our model are stated below:

$$\frac{dA}{dt} = -cA - v_Y\left(1 - e^{-\delta_A Y}\right)A \qquad (1)$$

$$\frac{dY}{dt} = -r_1 YC + r_2\left(1 - e^{-\delta_Y A}\right)\left(1 - \frac{Y}{Y_{max}}\right)Y \qquad (2)$$

$$\frac{dC}{dt} = r_3 v_C v_Y\left(1 - e^{-\delta_C A}\right)\left(1 - \frac{C}{C_{max}}\right)C \qquad (3)$$

where $v_C = kC(0)$, $C_{max} = k_C C(0)$ and $Y_{max} = k_Y(0)$.

Equation (1) determines the autophagy levels as a consequence of the cumulative effects of the primarily depletion rate (genetic or chemical inhibition) (first term) and the decrease in autophagosome formation rate caused by reduction in YAP/TAZ activity (second term). The effectiveness of the feedback effect caused by YAP/TAZ on autophagy levels is bounded, as we have previously seen[21]. More precisely, only strong YAP/TAZ depletion (corresponding to the case of high initial YAP and low output/final YAP) causes a major inhibition of autophagy, while a small reduction in YAP/TAZ levels (corresponding to the case of low initial YAP) barely impacts autophagy, similar to a plateau effect[21]. We therefore used a saturation term $(1-e^{-\delta_A Y})$, adapted from[44,45], to represent this described effect.

The strength of the feedback loop, which may vary between different cell types, was represented by the variable $v_Y$.

Equation (2) determines the YAP activity levels as the difference between the accumulation YAP/TAZ rate (second term) and the sequestration YAP/TAZ rate by α-catenin (first term) upon autophagy depletion. YAP activity follows a saturation curve of a maximum value of $Y_{max}$.

Equation (3) determines the α-catenin levels which accumulate upon autophagy inhibition (α-catenin is an LC3-interacting protein and a direct substrate of autophagy). The accumulation of α-catenin depends on the initial α-catenin levels (as we noticed a higher CTNNA1 protein levels after autophagy depletion in cell lines with high initial values—see Fig. 10a). α-Catenin protein levels follow a saturation curve of a maximum value of $C_{max}$. To simplify the model, the protein synthesis rates were neglected.

These three ODEs were numerically solved using MAPLE software. Specifically, the following values were used to define the constants included in this mathematical model $c = 0.1$, $r_1 = 1.0$, $r_2 = 1.2$, $r_3 = 3.0$, $k = 1.0$, $k_C = 3.0$, $k_Y = 4.0$ and $\delta_A = \delta_C = \delta_Y = 1.0$. At $t = 0$ (initial state, $t_0$), $A = 1.0$ and $Y = 1.0$, labelled as A_init and Y_init in the MAPLE code, respectively. The variables $v_Y$ (which define the strength of the feedback loop—how YAP/TAZ impacts on autophagy levels) and $v_C$ (which equals the α-catenin protein levels at time $t_0$, α-cat) were varied from 0.1 to 1.0 (number of steps: 10). The parameters A_init and Y_init were further swept from 0.1 to 2.0. The MAPLE code is included as Script.mw, together with step-by-step instructions (the files named README.md and Maple_Installation_guide.txt) and a typical output example (Data_Output.zip) in the GitHub repository https://github.com/rtuaic/Pavel-Nature-Communications-2021.

**Reporting summary.** Further information on research design is available in the Nature Research Reporting Summary linked to this article.

## Data availability
The authors can confirm that all relevant data are included in the paper and/or its Supplementary Information and Source Data files. The following databases were used in this study: iLIR database (https://ilir.warwick.ac.uk)[42], human Uniprot database (downloaded 03/06/14, 20,176 entries) and RCSB PDB (www.rcsb.org)[46]. CTNNA1 structure is available from PDB with Accession Number 4IGG [4]. The mass-spectrometry proteomics data have been deposited at the ProteomeXchange Consortium (http://proteomecentral.proteomexchange.org) via the PRIDE[47] partner repository with the dataset identifier P1. Source data are provided with this paper.

## Code availability
The MAPLE code used to generate the numerical data from Fig. 9e and Supplementary Figs. 22, 24–30 is included as Script.mw, together with step-by-step instructions (the files named README.md and Maple_Installation_guide.txt) and a typical output example (Data_Output.zip) in the GitHub repository https://github.com/rtuaic/Pavel-Nature-Communications-2021[48].

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

## Acknowledgements

We are very grateful to Dr Herbert Virgin for the Atg16L1 hypomorph mice and Dr. Tamotsu Yoshimori (Osaka University, Japan) for kindly providing us the mRFP-LC3 construct. We are also grateful to Dr. Folma Buss for kindly providing the A549 cells. We thank to Dr. Matthew Gratian and Dr. Mark Bowen (Cambridge Institute for Medical Research) for technical assistance. We also thank to Dr. Robin Antrobus and Dr. Nicholas J Matheson for their help in mass-spectrometry analysis of SILAC samples. This work was supported by the UK Dementia Research Institute (funded by the MRC, Alzheimer's Research UK and the Alzheimer's Society) (DCR) and The Roger de Spoelberch Foundation (DCR), Wellcome Trust [095317/Z/11/Z and 100140/Z/12/Z], a Romanian grant of Ministery of Research and Innovation CNCS–UEFISCDI, project number PN-III-P1-1.1-PD-2019-0733, within PNCDI III (MP) and L'Oréal-UNESCO For Women in Science Awards Programme (M.P.).

## Author contributions

M.P., S.J.P. and D.C.R. developed the rationale of the study and wrote the manuscript, which was commented on by all authors; M.P. and S.J.P. designed and performed most of the experiments with additional experimental contributions of C.F.B., F.M.M, T.R., R.A.F., M.R., S.M.S. and M.M.M., R.T. designed the mathematical model. D.C.R. supervised the study.

## Competing interests

F.M.M. is currently employed by Eli Lilly and C.F.B. is currently employed by Astex Pharmaceuticals. D.C.R is a consultant for Aladdin Healthcare Technologies SE and Nido Biosciences. All other authors declare no competing interests.
