## [Peer Review File · Nature Communications]

REVIEWER COMMENTS

Reviewer #1 (Remarks to the Author):

This manuscript shows that alpha-catenin is a target for degradation by the autophagy pathway. Furthermore, the authors show in some cell types that by reducing alpha-catenin levels, autophagy causes activation of YAP and numerous downstream consequences of YAP activation such as increased cell proliferation, stress fiber formation, and cell spreading. This is consistent with previous data showing that autophagy promotes YAP activation. However, in other cell types autophagy seems to inhibit YAP activity, and whether autophagy promotes or inhibits cancer proliferation is unclear. The authors propose that the differences in YAP regulation and cellular responses to autophagy are caused by varying levels of alpha-catenin in different cell types.

I found that the data showing that alpha-catenin was a new target of the autophagy pathway and a potential mediator of enhancement of YAP activity by autophagy compelling, interesting, and novel. Furthermore, if their model that varying alpha-catenin levels explain the different effects of autophagy on YAP regulation in different cell types is correct, it would also be an important and novel discovery. However I find the data supporting this claim confusing and in some cases contrary to the model. Since this is a major conclusion of the paper, I do not think the manuscript is acceptable for publication in its present form.

Main Points.

1) The primary concern that I have with the manuscript is with the claim that alpha-catenin levels dictate whether autophagy positively or negatively regulates YAP. First, the computational modeling of this phenomena is not well explained in the text. Given the regulatory input they describe (autophagy inhibits alpha-catenin, alpha-catenin inhibits YAP, and YAP promotes autophagy), I do not understand how autophagy inhibition causes increased YAP activity when alpha-catenin levels are low, without proposing autophagy also influences YAP activity through an additional mechanism. The authors need to walk through this step by step in the text to make it understandable to a broad audience. I am not saying that the computational model is wrong, it is just not obvious how it generates the results described.

More importantly, the authors also test their model by comparing the behavior of normal MCF10A cells to MCF10A with reduced levels of alpha-catenin. At the end of the Results it is stated: "As predicted by the model, autophagy inhibition in control MCF10A cells decreased nuclear YAP/TAZ and associated phenotypes like stress fibres and cell area, while autophagy inhibition in MCF10A cells depleted of CTNNA1/3 caused increased YAP/TAZ nuclear localisation, stress fibres and cell area (Fig. 7a-d)." While the first half of this sentence is correct, the second half is not. What the data in Figure 7b-d shows is that autophagy inhibition inhibits YAP activity in both normal and alpha-catenin depleted MCF10A cells, albeit to differing degrees. For example, if you compare YAP nuclear localization (Figure 7b) in alpha-catenin depleted cells, inhibition of autophagy (ATG7/10 KD) still causes reduction in nuclear YAP localization. The same pattern is true for other measures of YAP activity such as stress fibers and cell area (Figure 7c,d). Thus these experiments, which directly test their model, do not seem to support it.

Finally, their model predicts that cell lines with low alpha-catenin levels, “accumulate less α -catenins upon autophagy inhibition”. This prediction is tested in Figure 7g, but only a single cell line with low alpha-catenin is used. If this is truly a general behavior, then the authors should at least test the two other cell lines with low alpha-catenin used in Figure 7f.

Smaller points.

1) The first sentence of the Results needs a reference.

2) Figure 2 is supposed to show that treatments that stimulate autophagy cause increased YAP activity/nuclear localization. However from the images (Figure 2a,d,j), it appears that the treated cells are at lower cell density and thus would be predicted to have more YAP activity/nuclear localization even without inducing autophagy.

3) For Figure 4i, it is not clear what the labels on the horizontal axis stand for. This does not seem to be explained in the text or the legend.

4) For Figure 6a-c, it would be nice to see a control showing cells that had not been transfected.

Reviewer #2 (Remarks to the Author):

This manuscript describes a feedback pathway where autophagy can enhance Yap1/Taz signalling by downregulating the Yap/Taz inhibitor, α -catenin. (Or, put another way, Yap/Taz signalling is compromised when autophagy is inhibited, because α -catenin levels increase.) They identify α -catenin as an interaction partner of LC3 and show that this is functionally significant. Finally, they combine predictive modelling and experiment to show that pre-existing levels of α -catenin critically influence how this system behaves. The positive feedback described above occurs when α -catenin levels are relatively high, but is converted to a negative feedback pathway at lower α -catenin levels. This is an interesting test of the emergent properties of this signalling network.

Overall, this is a strong and exciting paper, that is supported by exhaustive experimentation (often replicated in multiple different cell lines – what, in an earlier age, we might have accepted just as “data not shown”, something that might make the paper easier to assimilate).

Specific comments.

1) I hesitate to say this, but there is one area that I think need further data (or a clarification of the data): Figure 7 a-d, which is the experimental test of the model. I think we need to see:

a) 7b is the key test of the model as a direct measure of Yap1 signaling. But the data need to be analysed statistically. The caption says that the experiment was “repeated with similar results”. But we need to know how many times and preferably shown a statistical comparison of the replicates.

b) This same stricture applies to Fig 7c.

2) Can the authors discuss, from their analysis of the model, why differences in a-catenin levels will generate qualitatively different behaviours from the system? I appreciate that this is speculative, but it will help generate intuitions for the reader and hypotheses for future work.

3) The association of a-catenin and LC3 has been studied by co-IP analyses. Whilst it is likely, the authors would need to perform binding studies with purified proteins before they can formally conclude that these proteins interact directly.

Peripheral questions – that interest me, but are not deal-breakers.

i) Fig 4e: LC3 interaction with a-catenin mutants. How many times has this been repeated?

ii) When you inhibit autophagy, where does the increase a-catenin go? Does any accumulate at junctions? Or is it all in the cytosol? This interests me, but is not a deal-breaker.

Reviewer #3 (Remarks to the Author):

Review of α -Catenin Levels Determine Direction of YAP/TAZ Response to Autophagy
2 Perturbation

Mariana Pavel, So Jung Park, Rebecca A. Frake, Carla F. Bento, Maurizio Renna,
Thomas Ricketts, Fiona M. Menzies, Sung Min Son, Radu Tanasa, & David C.
Rubinsztein

The authors consider the interaction between YAP/TAZ (Y), autophagy (A), and a proposed alpha-catenin (C). The authors had previously shown increasing Y, increases A via actin cytoskeleton genes. Here they perturb A and investigate the impact on Y. Decreasing autophagosome biogenesis yielded a lower N/C ratio of Y. Also, lower cell proliferation, manipulation of acini, and lysosome inhibition are also provided for evidence for lower A yielding lower Y activity. Going the other direction they also show that boosting A increases Y N/C ratio and downstream phenotypes associated with increased Y.

However, confounding data shows decreased A increases Y N/C ratio in liver cells, and was here replicated.

The hypothesis then is that the other cell types studied have a mechanism to counter A degradation of Y.

The authors identified a protein, alpha-catenin (C) that sequesters Y in the cytosol and thus acts to inhibit Y activity, and showed the presence of C in their non-liver cells as well as colocalization with an autophagy protein, LC3, and interaction with a specific domain, LIR motif.

This hypothesis is highly appropriate to test with a mathematical model to tease apart the quantitative nuance between the rise vs. the decline of Y with A.

I think the main argument is interesting and supported by the experiments. The model in Figs. 7e and Ext Data Fig 17 seem to support the rise of Y in the absence of C and the decline in Y with the presence of C as A declines in time.

So my only concerns are with the language around the model in relation to the output. I appreciate inclusion of the Maple code.

In the methods for Mathematical-numerical model

(i) says reduction in Y leads to reduction in A, but the 2nd term in the A equation, $-\nu_{\gamma}(1-\exp(-\delta_A Y))A$, suggests that as Y decreases toward zero, this term goes to zero leading to LESS loss of A, which I might interpret as NOT reducing A.

Similarly in

(ii) II) the claim is that there is an increase in Y activity when A is lower, but again the term $+(1-\exp(-\delta_{\gamma} A))$ is larger for LARGER A.

And in (iii)

since $dC/dt > 0$ C levels are always accumulating in some level of A, but depleting A would lower the rate of growth of C, suggesting A inhibits C production rather than promoting it. The authors mention Extended Data Fig 22 but those Figures only go up to 17, I believe.

The diagram (line 633) could be more illustrative. It suggests two mechanisms of A impact on Y but there is only the decrease of Y when A decreases in the 2nd term of the Y equation

Why are you tying parameters (e.g. ν_c , C_{max} , and Y_{max}) to the initial conditions? I understand nondimensionalizing the system but scaling by the initial conditions will bias the curves, though qualitatively the behavior should be similar. This is not the same as normalizing the curves.

Perhaps showing the model behavior as A grows as well would help to clarify my confusion.

Minor:

Do you show the liver cells (HepG2) do not express C?

The use of $\text{var}_{\{Y/T \rightarrow \text{autophagy}\}}$ in the text as a parameter when the model uses δ_A makes it harder to follow.

Parameter units are important and should be relayed. Some parameters in the text such as when $C > 0.5$ then Y decreases, while for $C < 0.3$, Y increases before declining. What does 0.5 or 0.3 mean?

Are you able to overexpress (or otherwise add in) C in a cell type with low baseline or initial C and suppress the Y activity. Figure 6 shows you are able to knockdown C in cell types with high C at baseline or initially and show drop in Y activity.

Reviewer #4 (Remarks to the Author):

Pavel et al. report that changes in the autophagy machinery in different cell types can cause contrasting effects in growth promoting oncogenic YAP/TAZ signalling pathway. Contrary to previous studies which stated that autophagy inhibition activates YAP/TAZ and promotes their nuclear localisation, this study performed in cells (MCF10A, HEK239, HeLa and pMECs) show that YAP/TAZ localisation is in-turn shifted to cytosol when autophagy is inhibited by inactivating ATG7/ATG10 or ATG16L1 and by treating cells with Bafilomycin A1. This study further reports that YAP/TAZ function is altered in cells with impaired autophagy using a range of cellular assays such as stress-fibre formation by monitoring F-actin staining, cell proliferation using BrdU incorporation, luciferase reporter assay using TEAD (a reporter of YAP/TAZ) and by assessing changes in spherical structure formations (acini) via growing cells in 3D stiff extracellular collagen matrix. Treating cells with autophagy activators showed increased YAP/TAZ activity. SILAC experiments in BafA1 treated HeLa cells identified α -catenin (CTNNA3) accumulation along with other autophagy substrates. The authors validated their interaction proteomics by showing that the protein levels of endogenous and overexpressed CTNNA3 and CTNNA1 increase and decrease under different autophagy inhibition and activation conditions, respectively. CTNNA3 was further shown to co-localise with LC3 and that this colocalization was increased under TAT-Beclin and BafA1 treated conditions.

Immunoprecipitation assays revealed an interaction of CTNNA1 with LC3. Putative LC3 interacting motifs were identified in-silico. Overexpression of CTNNA1 LIR mutants that possibly lost interaction with LC3 shows more altered YAP/TAZ nuclear localisation and functioning compared to wild-type CTNNA1. Dual knockdown of CTNNA1/3 and ATG7/ATG10 rescued the loss of YAP/TAZ functions observed upon individual ATG7/ATG10 knockdown, suggesting that CTNNA levels regulate YAP/TAZ functioning in an autophagy related manner. Finally, since CTNNA1 protein levels were found to be different in varied cell types and were increased upon ATG7/ATG10 knockdown in cells, the authors hypothesised that CTNNA acts as a autophagy substrate and prevents YAP/TAZ nuclear functioning by retaining it in cytoplasm when autophagy is inhibited. Overall, the work of Pavel and colleagues provide molecular insight into an intriguing cell identity determining seesaw-like signalling circuit. However, several concerns remain.

Major points:

1) Statistics missing in Figure 1b,g ; fig 2 b, c, e, k, l ; fig 6 c; fig 7 b, c. It is not clear if these experiments were done at least in triplicates.

2) In Figure 4 e, the Flag-trap blot LC3 higher exposure shows LC3 even in empty-flag. A better LC3 blot with no background binding is required to prove that the LC3B-CTNNA1 interaction is specific. Figure 4 f is missing a negative control (e.g. empty flag or even better overexpression of an unrelated Flag-tagged protein).

3) In Fig 4 h, the localisation of CTNNA1 mutant seemed to change compared to WT, what happens to localisation of other mutants and how does it change under different autophagy inhibition/activation conditions?

4) In Fig 5 d, the levels of CTNNA1 Y419-V422A is reduced by half (0.5) upon 50 μ M SMER28 treatment. This is in contrast with Figure 5 e where protein levels of this mutant does not change upon growth in EBSS.

5) Figure 6, a control with empty vector transfection should be included.

6) To confirm the findings from Figure 6 a in a more quantitative manner, the authors should perform subcellular fractionation and determine the protein amounts of endogenous CTNNA1 and YAP/TAZ in the nuclear and cytosolic fraction by immunoblotting.

7) Figure 7 a should be extended to CTNNA1/3 KD in autophagy activation conditions.

8) Does CTNNA1 binds YAP/TAZ in the cytosol? The authors should examine how this interaction is modified by CTNNA1 LTR mutants and by autophagy inhibited and activated conditions.

9) The author should provide evidence that CTNNA1 is delivered to the lysosome. This can be done by colocalization with a lysosomal marker (e.g. LAMP1) or distribution in a lysosomal fraction (e.g. subcellular fractionation or LysoIP).

10) To prove their model, the authors should re-tune CTNNA1 levels to rewire cell lines which show decreased YAP/TAZ signalling after autophagy induction into those in which autophagy induction promotes YAP/TAZ activity.

11) In Figure 1 a the authors use MCF10A cells to show that YAP/TAZ is in the nucleus in unperturbed conditions. However, in Figure 2 control-treated (unperturbed) MCF10A cells show YAP/TAZ in the cytosol. The authors should explain this discrepancy.

Minor points:

12) In Figure 4 g, it is not clear why only one of the two LIRs are shown. The authors should provide a similar figure assembly for the second LIR in CTNNA1.

13) Molecular weight markers are missing throughout the figures.

14) Please rephrase the first three sentences of the introduction as they are identical to the first three ones in the abstract.

Responses to reviewers

Reviewer #1

This manuscript shows that alpha-catenin is a target for degradation by the autophagy pathway. Furthermore, the authors show in some cell types that by reducing alpha-catenin levels, autophagy causes activation of YAP and numerous downstream consequences of YAP activation such as increased cell proliferation, stress fiber formation, and cell spreading. This is consistent with previous data showing that autophagy promotes YAP activation. However, in other cell types autophagy seems to inhibit YAP activity, and whether autophagy promotes or inhibits cancer proliferation is unclear. The authors propose that the differences in YAP regulation and cellular responses to autophagy are caused by varying levels of alpha-catenin in different cell types.

I found that the data showing that alpha-catenin was a new target of the autophagy pathway and a potential mediator of enhancement of YAP activity by autophagy compelling, interesting, and novel. Furthermore, if their model that varying alpha-catenin levels explain the different effects of autophagy on YAP regulation in different cell types is correct, it would also be an important and novel discovery. However I find the data supporting this claim confusing and in some cases contrary to the model. Since this is a major conclusion of the paper, I do not think the manuscript is acceptable for publication in its present form.

First we would like to thank the reviewer for the valuable comments and for appreciating the novelty, merit and importance of our work.

Comment 1:

Main Points.

1) The primary concern that I have with the manuscript is with the claim that alpha-catenin levels dictate whether autophagy positively or negatively regulates YAP. First, the computational modeling of this phenomena is not well explained in the text. Given the regulatory input they describe (autophagy inhibits alpha-catenin, alpha-catenin inhibits YAP, and YAP promotes autophagy), I do not understand how autophagy inhibition causes increased YAP activity when alpha-catenin levels are low, without proposing autophagy also influences YAP activity though an additional mechanism. The authors need to walk through this step by step in the text to make it understandable to a broad audience. I am not saying that the computational model is wrong, it is just not obvious how it generates the results described.

Answer 1:

First of all, we would like to apologize for not explaining clearly the 4 hypotheses involved in the generation of the model. Apart from the 3 hypothesis mentioned by the reviewer (autophagy inhibits alpha-catenin, alpha-catenin inhibits YAP, and YAP promotes autophagy), our model also takes into account a fourth hypothesis: autophagy inhibits YAP/TAZ, by promoting its degradation (the mechanism described by Lee et al. 2018¹⁵). Thus, in our mathematical model the final YAP/TAZ activity depends on the balance between 2 opposite mechanisms: a) the mechanism that is the focus of our current paper where autophagy inhibition compromises YAP activity (an indirect effect, via α -catenins), and b) the mechanism for increasing YAP upon autophagy inhibition (direct effect) described by Lee et al. 2018¹⁵.

These 4 hypothesis are explained in the **Methods section** → **Mathematical-Numerical Model subsection**, while describing the system of 3 differential equations (page 33, lines 738-752).

“The following basic assumptions were considered (see the scheme bellow):

- (i) Reduction in YAP/TAZ activity leads to autophagy decrease via previously described mechanisms^{20, 30}. When initial YAP levels are high, the extent of autophagy inhibition is also high, and when initial YAP levels are low, there is still a loss of autophagy, but the effects are small.
- (ii) YAP/TAZ activity results from the balance between two opposite mechanisms: I) *reduction in YAP/TAZ activity* as a consequence of YAP/TAZ being sequestered into the cytosol by α -catenin (the LIR-dependent mechanism experimentally described herein), and II) *increase in YAP/TAZ activity* as a consequence of perturbed autophagy (the mechanism experimentally described by Lee et al. 2018¹⁷) – when initial autophagy is high, the increase in YAP resulting from autophagy inhibition is large, and when initial autophagy is low, there is still an increase in YAP, but marginal.
- (iii) α -catenin protein levels accumulate upon autophagy perturbation (the mechanism experimentally described herein) – the rate of α -catenin accumulation is high when initial autophagy is high, and is low when autophagy is low.”

We have now extensively mentioned all 4 hypothesis governing the system of differential equations in the **Expended Data Fig. 28**. Indeed, **our model also considers the direct effect of autophagy upon YAP/TAZ** (mechanism previously published by Lee et al):

“The mathematical model proposed for understanding the dynamics of these 3 interlinked variables (autophagy, YAP activity and α -catenins) is based on the following **4 hypotheses**:

- (1) **YAP promotes autophagy** (YAP increase induces autophagy and YAP decrease inhibits autophagy – Pavel et. al. 2018)
- (2) **Autophagy promotes direct YAP inhibition** (autophagy induction directly decreases and inhibits YAP, **autophagy inhibition directly increases and activates YAP** – Lee et. al 2018)
- (3) **α -catenins inhibits YAP** (accumulation of α -catenins decreases YAP, α -catenin depletion and degradation increases YAP – Schlegelmilch et al. 2011)
- (4) **Autophagy degrades α -catenins** (*the novel mechanism described in this paper: LIR-dependent degradation of α -catenins*).

The description of the computational modelling has been expanded (see pages 33-34).

Comment 2:

More importantly, the authors also test their model by comparing the behavior of normal MCF10A cells to MCF10A with reduced levels of alpha-catenin. At the end of the Results it is stated: “As predicted by the model, autophagy inhibition in control MCF10A cells decreased nuclear YAP/TAZ and associated phenotypes like stress fibres and cell area, while autophagy inhibition in MCF10A cells depleted of CTNNA1/3 caused increased YAP/TAZ nuclear localisation, stress fibres and cell area (Fig. 7a-d).” While the first half of this sentence is correct, the second half is not. What the data in Figure 7b-d shows is that autophagy inhibition inhibits YAP activity in both normal and alpha-catenin depleted MCF10A cells, albeit to differing degrees. For example, if you compare YAP nuclear localization (Figure 7b) in alpha-catenin depleted cells, inhibition of autophagy (ATG7/10 KD) still causes reduction in nuclear YAP localization. The same pattern is true for other measures of YAP activity such as stress fibers and cell area (Figure 7c,d). Thus these experiments, which directly test their model, do not seem to support it.

Answer 2:

We would like to apologize again for not providing sufficient technical details regarding the performance of the respective experiments, previously found in **Figure 7a-d**, now included in **Figure 9a-d**.

This particular experiment was designed in order to give supportive evidence for the role of α -catenins in mediating the indirect effect of autophagy over YAP/TAZ activity. This experiment involved an initial knockdown of ATG7 and ATG10 for 48 hours (to cause autophagy inhibition) in MCF10A cells, and only after that, we depleted CTNNA1/3 KD for a further 48 hours. This experimental was intended to investigate if the effect of autophagy inhibition on reducing YAP/TAZ activity is largely mediated by the intracellular accumulation of α -catenins. Indeed, this experiment does not provide enough evidence regarding the role of initial levels of alpha-catenin (low vs high) in controlling the direction of YAP/TAZ response to autophagy perturbation, and is not the correct design to allow one to assess the effects of starting levels of alpha-catenins on YAP/TAZ, since the autophagy was inhibited first and the alpha-catenins were inhibited subsequently. We have now corrected the text on page 10, lines 217-225:

“To further confirm the role of α -catenins (CTNNA1, CTNNA3) as effectors for the YAP/TAZ phenotype caused by autophagy perturbation, MCF10A cells were initially exposed to ATG7/10 siRNAs for 48 hours in order to reach an autophagy-deficient status, and only after knockdowns, α -catenins were depleted using CTNNA1/3 siRNAs for other 48 hours. While ATG7/10-knockdown caused accumulation of α -catenins and subsequent YAP cytosolic sequestration (Fig. 9a-b) with consequent changes in the actin cytoskeleton morphology (Fig. 9c) and cell area (Fig. 9d) in MCF10A cells, the subsequent knockdown of α -catenins in the autophagy-compromised cells was, indeed, able to partially rescue those previously-perturbed YAP/TAZ-related phenotypes (Fig. 9a-d and Extended Data Fig. 21).”

In order to provide a compelling setup for investigating the role of initial alpha-catenin levels on final YAP/TAZ activity, we have designed a new experiment where we have tested the effects of reducing alpha-catenin levels in cells with normally high levels and overexpressing alpha-synuclein in cells with normally low levels to formally test our model.

See pages 13-14 and lines 281-306:

“Our modelling posits that starting levels of α -catenins determine the direction of the YAP/TAZ signalling response to autophagy perturbation (Fig. 9g, see also the extended explanation of the mathematical model in Extended Data Fig. 28), and can explain the divergent effects of autophagy perturbation on YAP/TAZ signalling in MCF10A cells (high starting levels of α -catenins) versus HepG2 cells (low starting levels of α -catenins). To further test the central role of α -catenin levels in this model, we have engineered MCF10A cells to have low levels of α -catenins (using siRNA knockdowns) and HepG2 cells to have high levels of α -catenins by overexpression (Fig 10 and Extended Data Figs. 29-30). In MCF10A cells previously depleted of α -catenins by siRNA knockdowns for 48 hours, untreated cells had higher YAP/TAZ activity than those exposed to autophagy stimuli with EBSS for 6 h (the experimental YAP/TAZ activity data is shown in Fig. 10b, while the corresponding modelling values are depicted in Extended Data Fig. 29a-c) or Trehalose 100 mM for 24 h (the experimental YAP/TAZ activity data is shown in Fig. 10c, while the corresponding modelling values are depicted in Extended Data Fig. 29d; the input YAP activity values in our modelling system were corresponded to those achieved experimentally by CTNNA1/3 knockdown under basal conditions). Conversely, a short-time exposure to ATG7/10 siRNAs (2 days) in MCF10A cells previously depleted of α -catenins for 48 hours, increased TEAD-YAP luciferase activity (the experimental YAP/TAZ activity data is shown in Fig. 10d, while the corresponding modelling values are depicted by the orange vertical bar on the left in Extended Data Fig. 30a) similar to cells characterised by initial low levels of α -catenins, such as HepG2. Consistent with our model, in HepG2 cells overexpressing the mEm-CTNNA1 construct, EBSS treatment for 6 h enhanced the YAP/TAZ activity in the HepG2 cells (the experimental YAP/TAZ activity data is shown in Fig. 10e, while the corresponding modelling values are depicted in Extended Data Fig. 29e). Thus, the HepG2 cells overexpressing wild-type CTNNA1 phenocopied what we had seen in MCF10A cells (Fig. 3f). Overall, these experimental results suggest that initial levels of α -catenins determine the dynamics of YAP/TAZ response to various autophagy perturbations.”

In order to explain why the timings are important (and why we have interpreted our experiments the way we have described above) we have expanded the text with new modelling and explanations (pages 14-15, lines 307-320):

“Since the autophagy perturbation is a dynamic process that spans over a certain time-period, one should also consider the time-dependency of autophagy perturbations on YAP/TAZ activity, detailed in Extended Data Figs.28,30. It is interesting to note that our modelling predicts that in low initial α -catenin conditions, YAP/TAZ activity will increase as autophagy is inhibited at early time points (YAP increases while autophagy decreases – Extended Data Fig. 30b, R correlation coefficient < -0.5). This effects is driven by the ability of autophagy to degrade YAP/TAZ^{17, 18} dominating the negative effect of impact of α -catenin accumulation on YAP/TAZ when α -catenin levels are low. However, when autophagy is inhibited then α -catenin also starts to increase and after a certain period of time α -catenin levels will reach a threshold when α -catenin sequestration of YAP/TAZ dominates the autophagy-mediated degradation of YAP/TAZ. From this timepoint onwards, YAP decreases as autophagy inhibition continues – Extended Data Fig. 30b,c, $R > 0.5$). In high initial α -catenin conditions, YAP/TAZ activity is predicted to positively correlate with autophagy levels at any relevant experimental time of the applied perturbation (Extended Data Figs. 24,26).”

We are grateful for your comment and suggestion as we now feel that our study is much more scientifically robust.

Comment 3:

Finally, their model predicts that cell lines with low alpha-catenin levels, “accumulate less α -catenins upon autophagy inhibition”. This prediction is tested in Figure 7g, but only a single cell line with low alpha-catenin is used. If this is truly a general behavior, then the authors should at least test the two other cell lines with low alpha-catenin used in Figure 7f.

Answer 3:

Please note that the editor instructed us that this experiment was not required.

However, we have now included new data that show the alpha-catenin levels after autophagy depletion in another cell line, Huh7 that has lower initial levels of CTNNA1 similar to the HepG2 cell-line - **Extended Data Fig. 23**. The increase in CTNNA1 levels are similar between these 2 cell-lines after ATG7/10 knockdown.

Please also see experiments described in the point above where we have tested the effects of reducing alpha-catenin levels in cells with normally high levels and overexpressing alpha-synuclein in cells with normally low levels to formally test our model.

Comment 4:

Smaller points.

1) The first sentence of the Results needs a reference.

Answer 4:

Thank you for your comment. We have now referenced our previous study: Pavel M, *et al.* Contact inhibition controls cell survival and proliferation via YAP/TAZ-autophagy axis. *Nature communications* **9**, 2961 (2018).

Comment 5:

2) Figure 2 is supposed to show that treatments that stimulate autophagy cause increased YAP activity/nuclear localization. However from the images (Figure 2a,d,j), it appears that the treated cells are at lower cell density and thus would be predicted to have more YAP activity/nuclear localization even without inducing autophagy.

Answer 5:

Autophagy induction increases cell size and might give the impression of lower cellular density, as fewer cells will be consequently imaged in a predefined microscopic area. Additionally, we have investigated the effect of autophagy induction in initially confluent cells in order to maximize the magnitude of the result. We have previously showed that when cells reach contact to each other, YAP/TAZ is predominantly localized into the cytosol, and this phenomenon is not potentiated by further increase in cell confluence (our previous paper from Nature Communications: Pavel *et al.* 2018, <https://www.nature.com/articles/s41467-018-05388-x> - Figure 1a-b, Supplementary Figure 3a). The F-actin staining from **Figs. 3-4** clearly shows that both untreated and autophagy-induced cells reached the confluent cellular state (where YAP/TAZ is expected to be sequestered in the cytosol under basal circumstances). We discussed this aspect at page 6, lines 124-126:

“Confluent MCF10A cells, with lower initial YAP/TAZ nuclear fractions²⁰, were used in these experiments in order to emphasize the YAP/TAZ activation mediated by autophagy induction.”

Comment 6:

3) For Figure 4i, it is not clear what the labels on the horizontal axis stand for. This does not seem to be explained in the text or the legend.

Answer 6:

Thank you for your comment. We have now explained the horizontal axis in the figure legend (**Fig. 6h**): The Pearson's (PC), Overlap (OC) and Mander's (M1 - fraction of LC3 overlapping mEm-CTNNA1, and M2 - fraction of mEm-CTNNA1 overlapping LC3) coefficients for MCF10A cells treated as in (G).

Comment 7:

4) For Figure 6a-c, it would be nice to see a control showing cells that had not been transfected.

Answer 7:

Thank you for this suggestion. We have now included the data of non-transfected control cells in **Extended Data Fig. 19**.

Reviewer #2

This manuscript describes a feedback pathway where autophagy can enhance Yap1/Taz signalling by downregulating the Yap/Taz inhibitor, α -catenin. (Or, put another way, Yap/Taz signalling is compromised when autophagy is inhibited, because α -catenin levels increase.) They identify α -catenin as an interaction partner of LC3 and show that this is functionally significant. Finally, they combine predictive modelling and experiment to show that pre-existing levels of α -catenin critically influence how this system behaves. The positive feedback described above occurs when α -catenin levels are relatively high, but is converted to a negative feedback pathway at lower α -catenin levels. This is an interesting test of the emergent properties of this signalling network.

Overall, this is a strong and exciting paper, that is supported by exhaustive experimentation (often replicated in multiple different cell lines – what, in an earlier age, we might have accepted just as “data not shown”, something that might make the paper easier to assimilate).

First, we would like to thank the reviewer for the valuable comments and we are very pleased to hear that our paper is strong and exciting and that the reviewer highly appreciated the amount of effort and work behind the present study.

Comment 1:

Specific comments.

1) I hesitate to say this, but there is one area that I think need further data (or a clarification of the data): Figure 7 a-d, which is the experimental test of the model. I think we need to see:

- a) 7b is the key test of the model as a direct measure of Yap1 signaling. But the data need to be analysed statistically. The caption says that the experiment was “repeated with similar results”. But we need to know how many times and preferably shown a statistical comparison of the replicates.
- b) This same stricture applies to Fig 7c.

Answer 1:

Thank you for your suggestion. We have now included the repeated experiments for previous **Fig. 7b-c** and **Extended Data Fig. 21a-d**. The pooled data from the 2 independent experiments were analyzed for statistical differences using the multinomial logistic regression. The significance of the observed differences in YAP/TAZ activity and percentage of cells with stress fibers between the 2 conditions, ATG7/10 KD and ATG7/10 KD + CTNNA1/3 KD, was analyzed by using the chi-squared test.

Please also see Answer 2 to reviewer 1 where in order to provide a compelling setup for investigating the role of initial alpha-catenins levels on final YAP/TAZ activity, we have designed a new experiment where we have tested the effects of reducing alpha-catenin levels in cells with normally high levels and overexpressing alpha-synuclein in cells with normally low levels to formally test our model.

Comment 2:

2) Can the authors discuss, from their analysis of the model, why differences in a-catenin levels will generate qualitatively different behaviours from the system? I appreciate that this is speculative, but it will help generate intuitions for the reader and hypotheses for future work.

Answer 2:

Please see new text on page 4, lines 66-71:

“Thus, we are showing that YAP/TAZ activity is a cellular readout that is determined by two competing processes: one being represented by the ability of autophagy to directly degrade YAP, and an opposite one being represented by the ability of autophagy to degrade α -catenins, which act as negative vectors for YAP/TAZ activity. Consequently, when α -catenins levels are low, autophagy inhibits YAP/TAZ activity, and when α -catenins levels are high, autophagy up-regulates YAP/TAZ activity.”

Please see new text (pages 14-15, lines 307-320):

“Since the autophagy perturbation is a dynamic process that spans over a certain time-period, one should also consider the time-dependency of autophagy perturbations on YAP/TAZ activity, detailed in Extended Data Figs.28,30. It is interesting to note that our modelling predicts that in low initial α -

catenin conditions, YAP/TAZ activity will increase as autophagy is inhibited at early time points (YAP increases while autophagy decreases – Extended Data Fig. 30b, R correlation coefficient < -0.5). This effect is driven by the ability of autophagy to degrade YAP/TAZ^{17, 18} dominating the negative effect of impact of α -catenin accumulation on YAP/TAZ when α -catenin levels are low. However, when autophagy is inhibited then α -catenin also starts to increase and after a certain period of time α -catenin levels will reach a threshold when α -catenin sequestration of YAP/TAZ dominates the autophagy-mediated degradation of YAP/TAZ. From this timepoint onwards, YAP decreases as autophagy inhibition continues – Extended Data Fig. 30b,c, $R > 0.5$). In high initial α -catenin conditions, YAP/TAZ activity is predicted to positively correlate with autophagy levels at any relevant experimental time of the applied perturbation (Extended Data Figs. 24,26).”

We have now also included a detailed explanation of the model and discussed the two initial situations of low and high alpha-catenin levels in new **Extended Data Figs. 28, 30**.

Extended Data Fig 28:

High initial C values.

When **initial C** values are **high**, the **initial tendency for YAP is to decrease** in differential equation (2), as the effect of the first term (corresponding to **hypothesis 3: autophagy inhibition indirectly inhibits YAP**) is higher than the effect provided by the second term (corresponding to **hypothesis 2: autophagy inhibition directly increases YAP**). As the *C* values increase over time, the strength of the first term also increases (more α -catenins in the cytosol, more YAP/TAZ sequestered and inactivated while autophagy decreases), while the rate of directly induced YAP accumulation by autophagy inhibition decreases over time (as this rate is proportional to the time-point YAP value). Thus, the **cumulative effect of autophagy inhibition over YAP is reduced YAP activity** (for the entire duration of the process).

Low initial C values.

When **initial C** values are **low**, the **initial tendency for YAP is to increase** in differential equation (2), as the effect of the first term (corresponding to **hypothesis 3: autophagy inhibition indirectly inhibits YAP**) is lower than the effect provided by the second term (corresponding to **hypothesis 2: autophagy inhibition directly increases YAP**). Thus, **an initial cumulative effect of autophagy inhibition over YAP is increased YAP activity**. However, as the *C* values increase over time, the strength of the first term also increases (more catenins in the cytosol, more YAP/TAZ sequestered and inactivated, while autophagy decreases), eventually reaching a time point when *the effect of the first term is equal to the second term* (corresponding to **the peak of the graph of YAP variation over time**, Extended Data Fig. 22a) and *even overcoming it*, determining YAP reduction over time if the system would be kept under such circumstances long enough, and, then, the **final possible cumulative effect of autophagy inhibition over YAP may be even the opposite: reduced YAP activity**.

and **Extended Data Fig. 30:**

Note: When initial α -cat values are low, the initial tendency for YAP is to increase as the direct effect of up-regulating YAP by autophagy inhibition (YAP accumulates due to its impaired autophagic degradation) overcomes the indirect effect of inhibiting YAP by autophagy depletion (the herein presented α -catenins-mediated mechanism). However, as the α -cat values accumulate over time in the autophagy-deficient cells, more YAP is sequestered into the cytosol and inactivated, as the indirect effect overcomes the previous direct effect. Thus, the cumulative effect at higher time points of autophagy inhibition is of reducing YAP activity.

Comment 3:

3) The association of α -catenin and LC3 has been studied by co-IP analyses. Whilst it is likely, the authors would need to perform binding studies with purified proteins before they can formally conclude that these proteins interact directly.

Answer 4:

The suggested experiment strengthen indeed the co-IP analyses. We have now performed in-vitro binding studies with purified proteins and included new data in **Extended Data Fig.12**. This experiment clearly shows the direct interaction between CTNNA1 and LC3.

We amended the main text on page 8, lines 164-167:

“The CTNNA1-LC3 interaction was further confirmed by *in vitro* binding assays (Extended Data Fig. 12) and by co-immunoprecipitation experiments between endogenous or overexpressed α -catenins and LC3 (Fig. 6d-f).”

Peripheral questions – that interest me, but are not deal-breakers.

i) Fig 4e: LC3 interaction with a-catenin mutants. How many times has this been repeated?

We investigated the interaction between LC3 and the alpha-catenin mutants 4 times for each of the experiments involving either Flag-tagged (**Extended Data Fig. 14b**) or mEm-tagged alpha-catenins (wt and mutants).

ii) When you inhibit autophagy, where does the increase a-catenin go? Does any accumulate at junctions? Or is it all in the cytosol? This interests me, but is not a deal-breaker.

For this study, we only focused on the localization of alpha-catenins in autophagosomes and autolysosomes under BafA1 condition. We obviously noticed an increased accumulation of catenins not only in autophagosomes/autolysosomes (**Figure 6h, Extended Data Figs. 13,15**), but also at the cell membrane.

Reviewer #3

The authors consider the interaction between YAP/TAZ (Y), autophagy (A), and a proposed alpha-catenin (C). The authors had previously shown increasing Y, increases A via actin cytoskeleton genes. Here they perturb A and investigate the impact on Y. Decreasing autophagosome biogenesis yielded a lower N/C ratio of Y. Also, lower cell proliferation, manipulation of acini, and lysosome inhibition are also provided for evidence for lower A yielding lower Y activity. Going the other direction they also show that boosting A increases Y N/C ratio and downstream phenotypes associated with increased Y. However, confounding data shows decreased A increases Y N/C ratio in liver cells, and was here replicated. The hypothesis then is that the other cell types studied have a mechanism to counter A degradation of Y.

The authors identified a protein, alpha-catenin (C) that sequesters Y in the cytosol and thus acts to inhibit Y activity, and showed the presence of C in their non-liver cells as well as colocalization with an autophagy protein, LC3, and interaction with a specific domain, LIR motif.

This hypothesis is highly appropriate to test with a mathematical model to tease apart the quantitative nuance between the rise vs. the decline of Y with A.

I think the main argument is interesting and supported by the experiments. The model in Figs. 7e and Ext Data Fig 17 seem to support the rise of Y in the absence of C and the decline in Y with the presence of C as A declines in time.

First, we would like to thank the reviewer for the valuable comments and we are pleased to hear that the main argument is interesting and supported by the experiments. We have now corrected the language around the mathematical-numerical model.

Comment 1:

So my only concerns are with the language around the model in relation to the output. I appreciate inclusion of the Maple code.

In the methods for Mathematical-numerical model

(i) says reduction in Y leads to reduction in A, but the 2nd term in the A equation, $-\nu_{\gamma}(1-\exp(-\delta_A Y))A$, suggests that as Y decreases toward zero, this term goes to zero leading to LESS loss of A, which I might interpret as NOT reducing A.

Answer 1:

Thank you for the suggestion and we apologise for not explaining clearly each term of the equation. We agree with your observation and thus, we emphasized it in the Method section, as follows (page 33, lines 742-744):

- (i) Reduction in YAP/TAZ activity leads to autophagy decrease via previously described mechanisms^{20, 30}. When initial YAP levels are high, the extent of autophagy inhibition is also high, and when initial YAP levels are low, there is still a loss of autophagy, but the effects are small.

Comment 2:

(ii) II) the claim is that there is an increase in Y activity when A is lower, but again the term $+(1-\exp(-\delta_{\gamma} A))$ is larger for LARGER A.

Answer 2:

As suggested by the reviewer, we amended the text including those observations as follows (page 33, lines 749-752):

- (ii) YAP/TAZ activity results from the balance between two opposite mechanisms: I) *reduction in YAP/TAZ activity* as a consequence of YAP/TAZ being sequestered into the cytosol by α -catenin (the LIR-dependent mechanism experimentally described herein), and II) *increase in YAP/TAZ activity* as a consequence of perturbed autophagy (the mechanism experimentally described by Lee et al. 2018¹⁷) – when initial autophagy is high, the increase in YAP resulting from autophagy inhibition is large, and when initial autophagy is low, there is still an increase in YAP, but marginal.

Comment 3:

And in (iii)

since $dC/dt > 0$ C levels are always accumulating in some level of A, but depleting A would lower the rate of growth of C, suggesting A inhibits C production rather than promoting it. The authors mention Extended Data Fig 22 but those Figures only go up to 17, I believe.

Answer 3:

We completely agree with the reviewer, and we expanded the text accordingly (page 34, lines 752-754):

- (iii) α -catenin protein levels accumulate upon autophagy perturbation (the mechanism experimentally described herein) – the rate of α -catenin accumulation is high when initial autophagy is high, and is low when autophagy is low.

Comment 4:

The diagram (line 633) could be more illustrative. It suggests two mechanisms of A impact on Y but there is only the decrease of Y when A decreases in the 2nd term of the Y equation.

Thank you for the suggestion. We have now included a more illustrative diagram in the Method section.

Also, we have inserted a detailed explanation of the biological meaning for the set of differential equations, accompanied by a similar model scheme in **Extended Data Fig. 28**.

Comment 5:

Why are you tying parameters (e.g. ν_c , C_{max} , and Y_{max}) to the initial conditions? I understand nondimensionalizing the system but scaling by the initial conditions will bias the curves, though qualitatively the behavior should be similar. This is not the same as normalizing the curves. Perhaps showing the model behavior as A grows as well would help to clarify my confusion.

Answer 5:

The reviewer makes a valuable comment regarding the importance of varying the initial parameters values in our numerical simulations, which we performed for Y_{init} and A_{init} and included the results in new **Extended Data Figs 24-27**. The C_{max} and Y_{max} parameters were fixed based on our experimental observations, as generally, the amount of protein accumulation upon autophagy perturbation is limited in cells, e.g. the accumulation of autophagy substrates in autophagy inhibition conditions usually reach a maximum of 2-4 times increase, as in the case of polyQ-htt, α -synuclein A53T, p62 or ATXN3 (Pavel et al, 2018; Pavel et al 2016 – Nature Communications). For this study, accumulation of α -catenins reaches a saturating level of a maximum of 3 times fold increase after prolonged autophagy inhibition in MCF10A cells (**Fig 5c,h**).

The variation of Y_{init} and A_{init} is commented in the main text on pages 12-13, lines 265-280:

“The model may also predict a time-dependent effect on YAP/TAZ activity when the initial YAP activity (Extended Data Figs. 24-25) or autophagy (Extended Data Figs. 26-27) levels are varied. Specifically, higher initial YAP values ($Y_{init} > 1$) are predicted to cause slightly smaller relative increases in YAP activity (YAP/Y_{init} – bottom graph in Extended Data Fig. 24) when compared to lower initial YAP conditions ($Y_{init} < 1$) at initial time points, but only in the cells with low initial α -cat values (< 0.3 , Extended Data Fig. 24-25). For the cases characterised by high basal α -cat values (> 0.5), the initial levels of YAP activity are not predicted to have major influences on the system outputs (Extended Data Figs. 24-25). When initial autophagy levels are varied (with $Y_{init} = 1$), the numerical model estimates that YAP/TAZ activity changes faster upon autophagy inhibition in systems characterised by basal lower autophagy ($A_{init} < 1$), compared to higher initial autophagy conditions ($A_{init} > 1$), and this effect is amplified when initial α -cat values are increased (Extended Data. Figs. 26-27). Thus, we may conclude from our numerical simulations, that the initial levels of

YAP activity appear to predominantly impact the magnitude of the output (YAP/TAZ activity), rather than influencing its direction (of increasing or decreasing) or its time period, while the initial autophagy levels mainly impact on the time-span of the output-effect.”

We are also showing the model behavior when autophagy is up-regulated. We have analysed the YAP/TAZ activity for a reasonable maximum autophagy fold increase (= 3), and compared these results (of numerical simulations – **Extended Data Fig. 29**) with the ones achieved experimentally – **Fig 10b,c,e** (pages 13-14, lines 281-306). Indeed, these new numerical simulations considerably strengthen the paper. The values of initial parameters seem not to influence the qualitative behavior of the process, however we updated the Ymax value (to 1.75) in order to be consistent with our experimental observations.

“Our modelling posits that starting levels of α -catenins determine the direction of the YAP/YAZ signalling response to autophagy perturbation (Fig. 9g, see also the extended explanation of the mathematical model in Extended Data Fig. 28), and can explain the divergent effects of autophagy perturbation on YAP/TAZ signalling in MCF10A cells (high starting levels of α -catenins) versus HepG2 cells (low starting levels of α -catenins). To further test the central role of α -catenin levels in this model, we have engineered MCF10A cells to have low levels of α -catenins (using siRNA knockdowns) and HepG2 cells to have high levels of α -catenins by overexpression (Fig 10 and Extended Data Figs. 29-30). In MCF10A cells previously depleted of α -catenins by siRNA knockdowns for 48 hours, untreated cells had higher YAP/TAZ activity than those exposed to autophagy stimuli with EBSS for 6 h (the experimental YAP/TAZ activity data is shown in Fig. 10b, while the corresponding modelling values are depicted in Extended Data Fig. 29a-c) or Trehalose 100 mM for 24 h (the experimental YAP/TAZ activity data is shown in Fig. 10c, while the corresponding modelling values are depicted in Extended Data Fig. 29d; the input YAP activity values in our modelling system were corresponded to those achieved experimentally by CTNNA1/3 knockdown under basal conditions). Conversely, a short-time exposure to ATG7/10 siRNAs (2 days) in MCF10A cells previously depleted of α -catenins for 48 hours, increased TEAD-YAP luciferase activity (the experimental YAP/TAZ activity data is shown in Fig. 10d, while the corresponding modelling values are depicted by the orange vertical bar on the left in Extended Data Fig. 30a) similar to cells characterised by initial low levels of α -catenins, such as HepG2. Consistent with our model, in HepG2 cells overexpressing the mEm-CTNNA1 construct, EBSS treatment for 6 h enhanced the YAP/TAZ activity in the HepG2 cells (the experimental YAP/TAZ activity data is shown in Fig. 10e, while the corresponding modelling values are depicted in Extended Data Fig. 29e). Thus, the HepG2 cells overexpressing wild-type CTNNA1 phenocopied what we had seen in MCF10A cells (Fig. 3f). Overall, these experimental results suggest that initial levels of α -catenins determine the dynamics of YAP/TAZ response to various autophagy perturbations.”

Minor:

Comment 6:

Do you show the liver cells (HepG2) do not express C?

Answer 6:

Liver cells such as HepG2 (**Figure 9f**) and Huh7 (**Extended Data Fig. 23**) express much less alpha-catenin than MCF10A or HEK293T cells (the relative ratio between basal CTNNA1 in HepG2 or Huh 7 and basal CTNNA1 levels in MCF10A cells is 0.3).

Comment 7:

The use of `var_{Y/T->autophagy}` in the text as a parameter when the model uses `delta_A` makes it harder to follow.

Answer 7:

We thank the reviewer for the comment and apologize for creating any confusion. We have now corrected the main text, including only the names of the variables used in the **Methods section** (page 11, lines 243-246):

“Among the parameters we varied, the initial values of intracellular α -catenins (α -cat) and the strength of the feedback loop (the variable which controls the YAP/TAZ influence over autophagy, v_Y) had the highest impact on the behaviour of the autophagy-YAP/TAZ loop (Fig. 9e and Extended Data Fig. 22a).”

Comment 8:

Parameter units are important and should be relayed. Some parameters in the text such as when $C > 0.5$ then Y decreases, while for $C < 0.3$, Y increases before declining. What does 0.5 or 0.3 mean?

Answer 8:

The C and Y parameters are expressed as arbitrary units, as C (α -cat) value corresponds to the relative cellular levels of α -catenins when compared to the standard MCF10A cells (eg. CTNNA1 densitometry in **Fig 9f**).

We amended the main text as follows, pages 11,-12 lines 246-252:

“However, the initial intracellular values of α -catenins had the strongest influence over the YAP/TAZ activity upon autophagy depletion in our mathematical model: higher initial α -cat values (>0.5 in Fig. 9e, corresponding to cells with basal α -catenins levels of at least 50 % of those found in MCF10A cells) promoted a reduction in YAP/TAZ activity, while small initial α -cat values (<0.3 in Fig. 9e, corresponding to cell lines with lower initial CTNNA1 levels compared to MCF10A cells by at least 70 %) promoted the opposite effect (of increased YAP/TAZ activity) when autophagy was primarily compromised.”

Comment 9:

Are you able to overexpress (or otherwise add in) C in a cell type with low baseline or initial C and suppress the Y activity. Figure 6 shows you are able to knockdown C in cell types with high C at baseline or initially and show drop in Y activity.

Thank you

Answer 9:

Thank you for your suggestion. We have now overexpressed CTNNA1 in HepG2 (cells with basal low α -catenin levels) and exposed them to EBSS. We were able to mimic the YAP/TAZ activity behaviour seen in MCF10A cells (**Fig. 10e**). Furthermore, we did the converse too – knocking-down α -catenins in MCF10A cells that normally have high levels of these protein (Fig 10). See details in Answer 5 to Reviewer 3.

Reviewer #4

Pavel et al. report that changes in the autophagy machinery in different cell types can cause contrasting effects in growth promoting oncogenic YAP/TAZ signalling pathway. Contrary to previous studies which stated that autophagy inhibition activates YAP/TAZ and promotes their nuclear localisation, this study performed in cells (MCF10A, HEK239, HeLa and pMECs) show that YAP/TAZ localisation is in-turn shifted to cytosol when autophagy is inhibited by inactivating ATG7/ATG10 or ATG16L1 and by treating cells with Bafilomycin A1. This study further reports that YAP/TAZ function is altered in cells with impaired autophagy using a range of cellular assays such as stress-fibre formation by monitoring F-actin staining, cell proliferation using BrdU incorporation,

luciferase reporter assay using TEAD (a reporter of YAP/TAZ) and by assessing changes in spherical structure formations (acini) via growing cells in 3D stiff extracellular collagen matrix. Treating cells with autophagy activators showed increased YAP/TAZ activity. SILAC experiments in BafA1 treated HeLa cells identified α -catenin (CTNNA3) accumulation along with other autophagy substrates. The authors validated their interaction proteomics by showing that the protein levels of endogenous and overexpressed CTNNA3 and CTNNA1 increase and decrease under different autophagy inhibition and activation conditions, respectively. CTNNA3 was further shown to co-localise with LC3 and that this colocalization was increased under TAT-Beclin and BafA1 treated conditions. Immunoprecipitation assays revealed an interaction of CTNNA1 with LC3. Putative LC3 interacting motifs were identified in-silico. Overexpression of CTNNA1 LIR mutants that possibly lost interaction with LC3 shows more altered YAP/TAZ nuclear localisation and functioning compared to wild-type CTNNA1. Dual knockdown of CTNNA1/3 and ATG7/ATG10 rescued the loss of YAP/TAZ functions observed upon individual ATG7/ATG10 knockdown, suggesting that CTNNA levels regulate YAP/TAZ functioning in an autophagy related manner. Finally, since CTNNA1 protein levels were found to be different in varied cell types and were increased upon ATG7/ATG10 knockdown in cells, the authors hypothesised that CTNNA acts as a autophagy substrate and prevents YAP/TAZ nuclear functioning by retaining it in cytoplasm when autophagy is inhibited. Overall, the work of Pavel and colleagues provide molecular insight into an intriguing cell identity determining seesaw-like signalling circuit. However, several concerns remain.

First, we would like to thank the reviewer for the very nice comments and valuable suggestions. We have now strengthened the experimental validation of the mathematical model and the role of initial alpha-catenin levels in directing the YAP/TAZ response to autophagy perturbations.

Major points:

Comment 1:

1) Statistics missing in Figure 1b,g ; fig 2 b, c, e, k, l ; fig 6 c; fig 7 b, c. It is not clear if these experiments were done at least in triplicates.

Answer 1:

We have now included the statistics for the indicated experiments and extended the figure legends with their corresponding relevant information. We have largely performed chi-squared test as statistical analysis for these individual experiments.

Comment 2:

2) In Figure 4 e, the Flag-trap blot LC3 higher exposure shows LC3 even in empty-flag. A better LC3 blot with no background binding is required to prove that the LC3B-CTNNA1 interaction is specific. Figure 4 f is missing a negative control (e.g. empty flag or even better overexpression of an unrelated Flag-tagged protein).

Answer 2:

We have now replaced the initial co-IP western-blot of FLAG-CTNNA1 with endogenous LC3 with a new one in **Figure 6e** and quantified the amount of LC3 co-immunoprecipitated with Flag-CTNNA1 WT and mutants from 4 independent experiments, please see **Extended Data Fig. 14**.

For **Figure 4f**, we would like to apologise for including with the first submission a cropped blot without the empty-mEm control for LC3-II and GFP, the control lane being initially visible only for GAPDH. We have now included the uncropped blot for all markers - see new **Fig. 6f**.

Comment 3:

3) In Fig 4 h, the localisation of CTNNA1 mutant seemed to change compared to WT, what happens to localisation of other mutants and how does it change under different autophagy inhibition/activation conditions?

Answer 3:

We noticed indeed a slight change in the cellular localisations of the mutants compared to WT alpha-catenin. We did not focus on investigating this particular aspect, as it was, in our opinion, beyond the scope of the present study. Thus, we prefer at this point not to emphasize this observation in here as we believe that more experiments are required in order to identify the exact cellular compartment where those mutants might accumulate and only after that, we may be able to extract some solid, validated conclusions. This observation may be the starting point for a separate in-depth study.

Comment 4:

4) In Fig 5 d, the levels of CTNNA1 Y419-V422A is reduced by half (0.5) upon 50 μ M SMER28 treatment. This is in contrast with Figure 5 e where protein levels of this mutant does not change upon growth in EBSS.

Answer 4:

This is indeed a very interesting observation raised by the reviewer. We have some unpublished data regarding some additional mechanisms for SMER28 in controlling various cellular pathways, so we would like at this stage not to disclose them, as other lab members are investigating this aspect and preparing a manuscript.

Comment 5:

5) Figure 6, a control with empty vector transfection should be included.

Answer 5:

We have now included the required data for the empty-vector transfected and non-transfected cells related to current **Figure 8**. These new data are found in **Extended Data Fig. 21**.

Comment 6:

6) To confirm the findings from Figure 6 a in a more quantitative manner, the authors should perform subcellular fractionation and determine the protein amounts of endogenous CTNNA1 and YAP/TAZ in the nuclear and cytosolic fraction by immunoblotting.

Answer 6:

As suggested by the reviewer, we reinforced the increase in YAP/TAZ cytosolic fraction in cells transfected with LIR-defective CTNNA1 mutants. We performed the nuclear/ cytosolic fractionation in cells transfected with either mEm-empty, -WT or the indicated LIR-mutants and quantified the YAP/TAZ nuclear/cytosolic ratio from 5 independent experiments, as shown in **Extended Data Fig. 18**.

Comment 7:

7) Figure 7a should be extended to CTNNA1/3 KD in autophagy activation conditions.

Answer 7:

We have performed the suggested experiments and included the results in a new main figure, **Figure 10**. We have now exposed CTNNA1/3 KD MCF10A cells to EBSS (**Fig. 10b**) or Trehalose (**Fig. 10c**) as autophagy-inducers, and, as our model predicted (**Extended Data Fig. 29c-d**), autophagy induction had the opposite effect as it had in the control MCF10A cells: reduced YAP/TAZ activity. Interesting,

HepG2 cells overexpressing WT alpha-catenin behaved similar to control MCF10A cells upon autophagy activation: increased YAP/TAZ activity (**Fig 10e, Extended Data Fig. 29e**).

The main text was amended accordingly on pages 13-14, lines 281-306:

“Our modelling posits that starting levels of α -catenins determine the direction of the YAP/YAZ signalling response to autophagy perturbation (Fig. 9g, see also the extended explanation of the mathematical model in Extended Data Fig. 28), and can explain the divergent effects of autophagy perturbation on YAP/TAZ signalling in MCF10A cells (high starting levels of α -catenins) versus HepG2 cells (low starting levels of α -catenins). To further test the central role of α -catenin levels in this model, we have engineered MCF10A cells to have low levels of α -catenins (using siRNA knockdowns) and HepG2 cells to have high levels of α -catenins by overexpression (Fig 10 and Extended Data Figs. 29-30). In MCF10A cells previously depleted of α -catenins by siRNA knockdowns for 48 hours, untreated cells had higher YAP/TAZ activity than those exposed to autophagy stimuli with EBSS for 6 h (the experimental YAP/TAZ activity data is shown in Fig. 10b, while the corresponding modelling values are depicted in Extended Data Fig. 29a-c) or Trehalose 100 mM for 24 h (the experimental YAP/TAZ activity data is shown in Fig. 10c, while the corresponding modelling values are depicted in Extended Data Fig. 29d; the input YAP activity values in our modelling system were corresponded to those achieved experimentally by CTNNA1/3 knockdown under basal conditions). Conversely, a short-time exposure to ATG7/10 siRNAs (2 days) in MCF10A cells previously depleted of α -catenins for 48 hours, increased TEAD-YAP luciferase activity (the experimental YAP/TAZ activity data is shown in Fig. 10d, while the corresponding modelling values are depicted by the orange vertical bar on the left in Extended Data Fig. 30a) similar to cells characterised by initial low levels of α -catenins, such as HepG2. Consistent with our model, in HepG2 cells overexpressing the mEm-CTNNA1 construct, EBSS treatment for 6 h enhanced the YAP/TAZ activity in the HepG2 cells (the experimental YAP/TAZ activity data is shown in Fig. 10e, while the corresponding modelling values are depicted in Extended Data Fig. 29e). Thus, the HepG2 cells overexpressing wild-type CTNNA1 phenocopied what we had seen in MCF10A cells (Fig. 3f). Overall, these experimental results suggest that initial levels of α -catenins determine the dynamics of YAP/TAZ response to various autophagy perturbations.”

Comment 8:

8) Does CTNNA1 binds YAP/TAZ in the cytosol? The authors should examine how this interaction is modified by CTNNA1 LIR mutants and by autophagy inhibited and activated conditions.

Answer 8:

As suggested, we investigated the co-IP of YAP/TAZ by flag-tagged CTNNA1 WT or LIR-defective mutants in cytosolic fractions of control, autophagy-inhibited (BafA1, 200nM, 16 h) or autophagy-induced (EBSS, 6 h) HEK293T cells (**Extended Data Fig. 20**). In control cells, the interaction between YAP/TAZ and LIR-defective CTNNA1 mutants increased compared to WT. Further, BafA1 only largely increased the interaction between the WT-CTNNA1 and YAP/TAZ (at least 2 times fold) and did not have this type of effect on the mutants. Compatible results were seen with autophagy induction – see pages 9-10, lines 202-210:

“The cytosolic YAP retention by LIR-defective α -catenins was confirmed by increased cytosolic co-immunoprecipitation of GFP-YAP with Flag-tagged CTNNA1 mutants when compared to the wild-type form (Extended Data Fig. 20a). BafA1 treatment doubled the amount of YAP co-immunoprecipitated with wild-type CTNNA1 (Extended Data Fig. 20a), while EBSS caused the opposite effect (of decreasing the interaction) – Extended Data Fig. 20b. Conversely, LIR-defective α -catenins did not show a similar behaviour of increased interactions with YAP upon BafA1 treatment, or reduced interactions upon exposure to EBSS (Extended Data Fig. 20). These data suggest a LIR/autophagy-dependent mechanism for YAP-cytosolic sequestration by α -catenins.”

Comment 9:

9) The authors should provide evidence that CTNNA1 is delivered to the lysosome. This can be done by colocalization with a lysosomal marker (e.g. LAMP1) or distribution in a lysosomal fraction (e.g. subcellular fractionation or LysoIP).

Answer 9:

Thank you for this suggestion. We now show colocalisation of endogenous LAMP1 and CTNNA1 in **Extended Data Fig. 15**. This colocalisation increased upon short treatment with BafA1 (400 nM, 4 h) that primarily impairs the lysosomal pH. See new added text on page 8, lines 171-174:

“As autophagosomes ultimately fuse with lysosomes, we next confirmed a significant increase in colocalisation of CTNNA3 with the lysosomal marker, LAMP1 upon BafA1 treatment, which increases lysosomal pH and compromises its degradative capacity (400 nM, 4 h – Extended Data Fig. 15).”

Comment 10:

10) To prove their model, the authors should re-tune CTNNA1 levels to rewire cell lines which show decreased YAP/TAZ signalling after autophagy induction into those in which autophagy induction promotes YAP/TAZ activity.

Answer 10:

Thank you for your suggestion. We have now overexpressed CTNNA1 in HepG2 (cells with basal low α -catenin levels) and exposed them to EBSS. We were able to mimick the YAP/TAZ activity behaviour seen in MCF10A cells (**Fig. 10e**). Furthermore, we did the converse too – knocking down alpha-catenins in MCF10A cells that normally have high levels of these protein (Fig 10). See details in Answer 5 to Reviewer 3.

Comment 11:

11) In Figure 1a the authors use MCF10A cells to show that YAP/TAZ is in the nucleus in unperturbed conditions. However, in Figure 2 control-treated (unperturbed) MCF10A cells show YAP/TAZ in the cytosol. The authors should explain this discrepancy.

Answer 11:

YAP/TAZ shows a predominant nuclear localization in low confluency cells, while when cells reach the confluency state, YAP/TAZ goes into the cytosol. In order to be able to emphasize the effect of increasing YAP/TAZ activity upon autophagy up-regulation, we designed those experiments using confluent MCF10A cells which had initial lower YAP/TAZ nuclear fraction. Conversely, for showing the effect of sequestering YAP/TAZ into the cytosol by autophagy inhibition, the cells were rather studied at lower confluence, with initial YAP/TAZ being predominantly located into the nucleus. We used these experimental setups with the aim of increasing the magnitude of the autophagy effects over YAP/TAZ signaling.

We amended the text accordingly, page 6, lines 124-126:

“Confluent MCF10A cells, with lower initial YAP/TAZ nuclear fractions²⁰, were used in these experiments in order to emphasize the YAP/TAZ activation mediated by autophagy induction.”

Minor points:

Comment 12:

12) In Figure 4 g, it is not clear why only one of the two LIRs are shown. The authors should provide a similar figure assembly for the second LIR in CTNNA1.

Answer 12:

We have now included a similar figure assembly for the second LIR in CTNNA1 in **Extended Data Fig. 17a** (in the middle).

Comment 13:

13) Molecular weight markers are missing throughout the figures.

Answer 13:

We have now added the molecular weight markers.

Comment 14:

14) Please rephrase the first three sentences of the introduction as they are identical to the first three ones in the abstract.

Answer 14:

We rephrased the first few sentences and included the relevant references (pages 2-3, lines 38-44):

“The identity and status of cells is commonly depicted using static “omics” profiles: epigenomic, transcriptomic, proteomic or metabolomic^{1, 2, 3}. Changes in cellular identity play an important role in promoting diseases, including cancers, inflammatory conditions and neurodegeneration, and identifying the factors regulating these cell identity switches might provide novel therapeutic opportunities^{4, 5}. However, it is not well understood how such static hallmarks change over time and convert into dynamic responses to external or internal perturbations that ultimately determine cellular functions.”

REVIEWERS' COMMENTS

Reviewer #1 (Remarks to the Author):

The authors have done considerable work to address the concerns of the reviewers. I think that the additional experiments and explanations fully address my primary questions. However, I still have a couple suggestions that can be easily addressed that would improve the manuscript. The first is that the last few sentences of the results, just prior to the Discussion could be moved much earlier to introduce at a conceptual level how autophagy inhibition could have opposite effects on YAP activity depending in the initial levels of alpha-catenin. This information could be conveyed as an hypothesis to explain why the differing effects of autophagy inhibition on YAP could depend on alpha-catenin levels. The modeling and subsequent experiments are used to test the hypothesis. It is not critical that the model gets introduced in that manner, but bringing it up early would make the data more understandable and less confusing.

The other point I have is that the following sentence in the abstract is confusing: "High basal levels of α -catenins enable autophagy to positively regulate YAP/TAZ, while low α -catenins cause YAP/TAZ activation upon autophagy inhibition." I think it needs to start "High basal levels of α -catenins enable autophagy inhibition to positively regulate YAP/TAZ,....."

Reviewer #2 (Remarks to the Author):

The authors have done an extensive revision of their manuscript, which address all the issues that I raised in my earlier comments.

Reviewer #3 (Remarks to the Author):

Review of Resubmission of
 α -Catenin Levels Determine Direction of YAP/TAZ Response to Autophagy Perturbation

The authors have added considerable work and effort in addressing the reviewer comments. By and large I am satisfied with the additions and clarifications. The additional experiments and corresponding simulations are excellent. The explanation and simulation involving the use of the initial conditions within the model parameters is ok. The extensive simulations over various parameters in the extended figures are a nice addition if maybe a bit more than necessary.

Minor comments:

I like the addition of the figure in the model methods section. I would recommend altering the figure to include inhibition (flat) arrow heads for alpha-catenin's effect on YAP/TAZ and YAP/TAZ effects on Autophagy (similar to the flat arrowhead usage in Fig 9e).

--| A ___
| |
| | |
| V |
| |
| C |
| |
| | |
| - |
| |
---- Y <--

I do not feel the Maple code in the article itself is necessary.

Reviewer #4 (Remarks to the Author):

The authors have done an impressive job adequately and comprehensively addressing all my critical points. I therefore recommend to accept this manuscript for publication. Well done!

Responses to reviewers

Reviewer #1

The authors have done considerable work to address the concerns of the reviewers. I think that the additional experiments and explanations fully address my primary questions. However, I still have a couple suggestions that can be easily addressed that would improve the manuscript. The first is that the last few sentences of the results, just prior to the Discussion could be moved much earlier to introduce at a conceptual level how autophagy inhibition could have opposite effects on YAP activity depending in the initial levels of alpha-catenin. This information could be conveyed as an hypothesis to explain why the differing effects of autophagy inhibition on YAP could depend on alpha-catenin levels. The modeling and subsequent experiments are used to test the hypothesis. It is not critical that the model gets introduced in that manner, but bringing it up early would make the data more understandable and less confusing.

As suggested, we have now added the following sentences in the main file (page 11):

“As we described previously that YAP/TAZ positively regulate autophagosome biogenesis²⁰ and our current data suggest that α -catenins (known inhibitors of YAP/TAZ) are autophagy substrates, we considered whether a YAP/TAZ-autophagy-YAP/TAZ feedback loop operates in cells, with starting the levels of α -catenins explaining the output differences among various cellular systems. As autophagy is a dynamic process and we are postulating a feedback loop, we have considered the possibility that one might see different effects on YAP/TAZ activity at various times after initiating the autophagy perturbation. To explore this possibility, we created a numerical mathematical model based on 3 differential equations to measure the dynamics of autophagy levels, YAP/TAZ activity and levels of α -catenins, when autophagy is primarily impaired.”

The other point I have is that the following sentence in the abstract is confusing: "High basal levels of α -catenins enable autophagy to positively regulate YAP/TAZ, while low α -catenins cause YAP/TAZ activation upon autophagy inhibition." I think it needs to start "High basal levels of α -catenins enable autophagy inhibition to positively regulate YAP/TAZ,....."

We have now rephrased the abstract at follows:

“High basal levels of α -catenins enable autophagy induction to positively regulate YAP/TAZ, while low α -catenins cause YAP/TAZ activation upon autophagy inhibition.”

Reviewer #3

Review of Resubmission of

α -Catenin Levels Determine Direction of YAP/TAZ Response to Autophagy Perturbation

The authors have added considerable work and effort in addressing the reviewer comments. By and large I am satisfied with the additions and clarifications. The additional experiments and corresponding simulations are excellent. The explanation and simulation involving the use of the initial conditions within the model parameters is ok. The extensive simulations over various parameters in the extended figures are a nice addition if maybe a bit more than necessary.

Minor comments:

I like the addition of the figure in the model methods section. I would recommend altering the figure to include inhibition (flat) arrow heads for alpha-catenin's effect on YAP/TAZ and YAP/TAZ effects on Autophagy (similar to the flat arrowhead usage in Fig 9e).

```
--| A ___  
||  
||  
|V|  
||  
|C|  
||  
||  
|-|  
||  
---- Y <--
```

I do not feel the Maple code in the article itself is necessary.

First, we would like to thank the reviewer for the valuable comments. We have now corrected the figure as suggested (by including the flat arrow) and removed the MAPLE code from the Methods section. The MAPLE code is now included as Script.mw, together with step-by-step instructions (the files named README.md and Maple_Installation_guide.txt) and a typical output example (Data_Output.zip) in the GitHub repository <https://github.com/rтуаic/Pavel-Nature-Communications-2021> (in the Code availability section).